# Constrained Pareto Set Identification with Bandit Feedback

Cyrille Kone [1]   Emilie Kaufmann [1]   Laura Richert [2]

## Abstract

In this paper, we address the problem of identifying the Pareto Set under feasibility constraints in a multivariate bandit setting. Specifically, given a $K$-armed bandit with unknown means $\mu_1, \ldots, \mu_K \in \mathbb{R}^d$, the goal is to identify the set of arms whose mean is not uniformly worse than that of another arm (i.e., not smaller for all objectives), while satisfying some known set of linear constraints, expressing, for example, some minimal performance on each objective. Our focus lies in fixed-confidence identification, for which we introduce an algorithm that significantly outperforms racing-like algorithms and the intuitive two-stage approach that first identifies feasible arms and then their Pareto Set. We further prove an information-theoretic lower bound on the sample complexity of any algorithm for constrained Pareto Set identification, showing that the sample complexity of our approach is near-optimal. Our theoretical results are supported by an extensive empirical evaluation on a series of benchmarks.

## 1. Introduction

Pareto Set Identification (PSI) is a fundamental active testing problem in which a learner sequentially queries samples from a multi-variate multi-armed bandit to identify – as efficiently as possible – the arms whose average return satisfies optimal trade-offs among possibly competing objectives (Zuluaga et al., 2013; Auer et al., 2016). In this framework, a $K$-armed bandit is denoted $\nu := (\nu_1, \ldots, \nu_K)$, where each arm $\nu_k$ is a multi-variate distribution with mean $\mu_k \in \mathbb{R}^d$ and $d \geqslant 1$. Given a bandit model with means $\mu := (\mu_1, \ldots, \mu_K)$, its Pareto Set is the set of arms $k$ such that for any other arm $k'$, there exists an ob-

jective (or a marginal) $c$ for which arm $k$ is in expectation preferable to $k'$ (i.e. $\mu_k^c \geqslant \mu_{k'}^c$). Each arm in the Pareto Set satisfies an optimal trade-off between different objectives in the sense that no other arm can improve over it for all objectives. This setting extends the well-studied *Best Arm Identification* framework (e.g., Even-Dar et al. (2006); Audibert & Bubeck (2010); Jamieson & Nowak (2014)) to problems where multiple metrics need to be optimized simultaneously with no clear preference.

PSI has diverse applications, including recommender systems where multiple performance metrics are considered (Lin et al., 2019; Huang et al., 2019; Mehrotra et al., 2020), or hardware or software design optimization (Almer et al., 2011; Zuluaga et al., 2013), which frequently involves trade-offs between metrics such as energy consumption and performance. We are particularly interested in potential applications to clinical trials, in which multiple biological markers may be tracked to assess the efficacy of a treatment (Munro et al., 2021), or in which efficacy and safety may be jointly monitored (Jennison & Turnbull, 1993).

In the literature on PSI, no particular constraints are imposed on the arms in the Pareto Set, allowing for situations where an arm may exhibit very low expected performance on one objective but still be in the Pareto Set due to its good performance on another. This lack of constraints may be undesirable in practice. For instance, in clinical trials, it may be important to identify good candidate drugs having some minimal performance level on some indicator of efficacy (e.g., a minimal antibody response in the case of vaccine development) or avoiding exceeding some maximal toxicity threshold (Mark C et al., 2013). Similarly, in multi-objective recommender systems or material design optimization, one might be interested in finding the Pareto Set of items that satisfy some minimal performance metrics or dimensions constraints (Kumar et al., 2021; Afshari et al., 2019). Existing PSI algorithms are unable to handle such constraints. To fill this gap, we introduce a novel active testing problem termed *constrained Pareto Set Identification*. In this setting, the learner is given a polyhedron $P := \{x \in \mathbb{R}^d \mid \boldsymbol{A}x \leqslant b\}$ describing solutions to a set of linear constraints with $\boldsymbol{A} \in \mathbb{R}^{m,d}$ and $b \in \mathbb{R}^m$ (i.e., $m$ linear constraints). Any arm $k$ such that $\mu_k \in P$ is called feasible. The goal of the learner is to identify the Pareto Set of the feasible arms.

---

[1]Univ. Lille, Inria, CNRS, Centrale Lille, UMR 9198-CRIStAL, F-59000 Lille, France [2]Univ. Bordeaux, Inserm, Inria, BPH, U1219, Sistm, F-33000 Bordeaux, France. Correspondence to: Cyrille Kone <cyrille.kone@inria.fr>.

*Proceedings of the 42nd International Conference on Machine Learning*, Vancouver, Canada. PMLR 267, 2025. Copyright 2025 by the author(s).

We study the constrained PSI problem in the fixed-confidence setting. In this framework, the learner is allowed an unrestricted exploration phase and aims to identify the Pareto Set of the feasible arms with high confidence, while minimizing the expected length of the exploration phase, called the *sample complexity*.

## 1.1. Setting

We denote by $K$ the number of arms, $d \geqslant 1$ the number of objectives, and $[K] := \{1, \ldots, K\}$ the set of arms. We let $\mathcal{D}$ be a family of distributions in $\mathbb{R}^d$. Given a multivariate bandit instance $\nu := (\nu_1, \ldots, \nu_K) \in \mathcal{D}^K$, we denote the collection of its mean vectors by $\boldsymbol{\mu} := (\mu_1, \ldots, \mu_K) \in \mathcal{I}^K$, where $\mathcal{I} \subset \mathbb{R}^d$ and $\mu_k := \mathbb{E}_{X \sim \nu_k}[X]$. We shall use $\boldsymbol{\mu}$ to represent the distribution $\nu$. In the sequel $\mathcal{D}$ will be either a set of subGaussian distributions[1] (for algorithms) or a parametric set of multivariate Gaussian distribution (for which we will derive lower bounds).

At each time $t = 1, 2, \ldots$, the agent chooses an arm $k_t \in [K]$ (based on past observations) and observes an outcome $X_t \sim \eta_{k_t}$ independent of past actions and observations. With $\mathcal{F}_t := \sigma(k_1, X_1, \ldots, k_t, X_t, k_{t+1})$ the $\sigma$-algebra representing the history collected up to time $t$, we let $\mathcal{F} := \{\mathcal{F}_t, t \geqslant 1\}$ be the natural filtration associated to the sequential process. An algorithm for a multi-objective pure exploration problem –that, in general, aims at finding the answer to some question about the means $\boldsymbol{\mu}$ – is made of (1) a *sampling rule* to choose $k_t$ that is predictable i.e $k_{t+1}$ is $\mathcal{F}_t$-measurable; (2) a *recommendation rule* $R_t$ that gives an $\mathcal{F}_t$-measurable guess of the answer and finally (3) a *stopping rule* that determines a time $\tau$ to stop the data acquisition process, which is a stopping time w.r.t. to $\mathcal{F}$. The output of the algorithm is the recommendation made upon stopping, $R_\tau$. Given some algorithm, we denote by $\mathbb{P}_\nu$ the law of the stochastic process $\{k_t, X_t : t \geqslant 1\}$ and by $\mathbb{E}_\nu[\cdot]$, the expectation under $\mathbb{P}_\nu$. The subscript may be omitted when $\nu$ is clear from the context.

In the particular pure-exploration problems that are our focus in the paper, the agent is given a convex polyhedron $P := \{x \in \mathbb{R}^d \mid \boldsymbol{A}x \leqslant b\}$ where $\boldsymbol{A} \in \mathbb{R}^{m \times d}$ is matrix, $b \in \mathbb{R}^m$ and $m$ is the number of constraints. Given this polyhedron, we consider two learning tasks in this work.

**Constrained PSI (cPSI)** The learner should identify $O^*$, the Pareto Set of the arms that belong to $P$.

**Constrained PSI with explainability (e-cPSI)** The learner should identify $O^*$ but in addition she is asked to

provide a reason for rejecting any arm that does not belong to $O^*$. Every arm outside of $O^*$ should be classified as either infeasible (does not satisfy the constraints) or dominated by a feasible arm. Note that for arms that are both infeasible and dominated, both explanations are valid.

While cPSI may be adequate for certain applications, e-cPSI becomes particularly relevant in scenarios like clinical trials, where the experimenter needs a clear justification for rejecting each arm. This need for explainability makes cPSI especially important, and as a result, developing an efficient algorithm for it will be our primary focus.

We remark that Katz-Samuels & Scott (2019) studied a similar "$\delta$-PAC-EXPLANATORY" framework for the identification of a feasible arm that maximizes some linear combination of the objectives.

## 1.2. Contributions

A natural approach to constrained PSI consists in applying first an algorithm for identifying the feasible set (e.g., the one of Katz-Samuels & Scott (2018)) and then applying an algorithm to identify the Pareto set of the arms that were returned as feasible by the previous algorithm (e.g., that of Crepon et al. (2024)). We will see in Section 4 that such a naive two-staged approach can be arbitrary inefficient. This highlights the necessity to propose algorithms specifically tailored to constrained Pareto Set Identification, be it with or without the explainability component.

Our contributions are the following. First, we present in Section 3 sample complexity lower bounds for both the cPSI and e-cPSI problems. For cPSI, leveraging a gradient-ascent approach, we propose an algorithm matching the lower bound in the asymptotic regime in which the error probability goes to zero (see Algorithm 2 in Appendix B), with a complexity that remains polynomial in the number of arms. Yet such an approach cannot be used for the more challenging e-cPSI and we devote the rest of our effort to proposing and analyzing an efficient algorithm for this setting. In Section 4, we present e-cAPE, an algorithm leveraging upper and lower confidence bounds to efficiently balance exploration between feasibility detection and Pareto Set Identification. We provide an upper bound on its sample complexity and prove that it can be arbitrarily smaller than that of the two-stage baseline discussed above, and can be near-optimal for some classes of instances. Finally, in Section 5 we validate our algorithms through an empirical evaluation on two real-world datasets from clinical trials. Additional experiments are provided in Appendix G.

**Notation** Given $x \in \mathbb{R}^d$, let $x^c$ denotes its $c$-th component and $\mathrm{dist}(x, \mathcal{X}) := \inf\{\|x - y\|_2 : y \in \mathcal{X}\}$ the distance to $\mathcal{X}$, for $\mathcal{X} \subset \mathbb{R}^d$. $\mathrm{cl}(P)$ denotes the closure of $P$, $\mathrm{int}(P)$ its interior and $\partial P := \mathrm{cl}(P) \backslash \mathrm{int}(P)$, the boundary of $P$.

---

[1]Letting $(X_i^c)_{c \leqslant d}$ be a realization of arm $i$, it is marginally $\sigma$-subGaussian if for all $c \leqslant d$, $\forall \lambda \in \mathbb{R}$, $\mathbb{E}[e^{\lambda(X_i^c - \mu_i^c)}] \leqslant e^{\frac{\lambda^2 \sigma^2}{2}}$ and norm-$\sigma_u$-subGaussian if $\forall z \in \mathbb{R}^d$ such that $\|z\|_2 = 1$, $X_i^\intercal z$ is $\sigma_u$-subGaussian.

For $a, b \in \mathbb{R}$, $(a)_+ := \max(a, 0)$, $a \wedge b = \min(a, b)$ and $a \vee b = \max(a, b)$. We use bold symbols for matrices. A glossary of notation is provided in Appendix A.

## 2. Related Work

Best Arm Identification (BAI) can be seen as a special case of PSI when $d = 1$. In recent years, significant attention has been given to developing asymptotically optimal algorithms for fixed-confidence BAI when the arms distribution comes from a parametric family. These algorithms satisfy strong optimality guarantees on the sample complexity in the regime where the error probability $\delta$ is small. Notable contributions to this line of work include (Garivier & Kaufmann, 2016; Degenne & Koolen, 2019; Wang et al., 2021; You et al., 2023). Another line of research has developed efficient BAI algorithms using upper and lower confidence bounds (Kalyanakrishnan et al., 2012; Jamieson & Nowak, 2014), whose sample complexity bounds are expressed in terms of sub-optimality gaps, but is not matching the existing lower bound when $\delta$ goes to zero.

Extending these techniques to multi-variate bandits induces additional complexity due to the lack of a total ordering among arms. Recent work has successfully addressed these challenges by proposing efficient algorithms for PSI.

Zuluaga et al. (2013; 2016) studied PSI in the bandit setting and introduced an algorithm based on Gaussian Process regression techniques and confidence regions to adaptively identify the Pareto Set while sequentially discarding sub-optimal arms (i.e., arms that are not in the Pareto Set). Auer et al. (2016) introduced the first $(\varepsilon, \delta)$-PAC algorithm for PSI and proposed a lower bound on the sample complexity of this problem. The authors identified a problem-dependent complexity term $H(\boldsymbol{\mu})$ that is written as a sum of inverse square "sub-optimality gaps" and which is the leading term of the sample complexity. Karagözlü et al. (2024) further extended the algorithm of Auer et al. (2016) to bandit multi-objective optimization under any cone-induced pre-order. Kone et al. (2023) introduced a generalization of the LUCB algorithm (Kalyanakrishnan et al., 2012) for PSI and studied different relaxations of the problem to address different use cases. The algorithms of Auer et al. (2016) and Kone et al. (2023) both identify the Pareto Set by collecting at most $O\left(H(\boldsymbol{\mu})\log(K\log(H(\boldsymbol{\mu}))/\delta)\right)$ samples. An extension of PSI in which each arm is characterized by observable features on which its mean depends linearly was further studied by Kone et al. (2024b).

Crepon et al. (2024) proposed an algorithm with tight guarantees in the asymptotic regime (i.e., when $\delta \to 0$). Their algorithm is based on iteratively solving an optimization problem that characterizes the optimal problem-dependent sample complexity of PSI. The authors extended the online gradient-ascent algorithm of Ménard (2019) and proposed an efficient procedure for computing the (super)gradient of the optimization problem. Still in the same confidence regime, Kone et al. (2024a) introduced an algorithm that relies on the posterior distribution to determine which arm to sample at each round and when to stop collecting new samples. While these algorithms are optimal in the asymptotic regime for multivariate Gaussian arms, they do not come with an explicit upper bound on their sample complexity for a fixed confidence level $\delta$. Hence, their performance in the moderate confidence regime is still to be investigated. However, their practical performance is appealing even for moderate values of $\delta$ and subGaussian arms.

Notably, none of these works address the additional challenge of handling constraints on the Pareto Set.

Feasibility detection in multi-objective bandit problems was studied by Katz-Samuels & Scott (2018): given a set of linear constraints materialized by a polyhedron $P \subset \mathbb{R}^d$, one should identify the set of arms whose means belong to $P$. A constrained BAI problem was further studied by Katz-Samuels & Scott (2019): given a weight vector $w \in \mathbb{R}^d$, the learner should identify the arm that maximizes $w^\intercal \mu_k$ among arms $k$ belonging to $P$. While this approach assumes a known preference weighting, it fundamentally differs from constrained PSI, where no such preference vector $w$ exists to rank the feasible arms. In situations where there is no clear preference vector $w$, it may be desirable to find instead their Pareto Set. For example in early-stage vaccine research, several immune markers are of interest, and it is at that stage not yet known which one(s) are the most relevant to protect against new diseases.

Finally, while we consider in this work the identification of the Pareto Set under known constraints materialized by $P$, other works have studied the constrained regret minimization setting where the learner should pull arms with the largest mean (single-objective, see Lattimore & Szepesvári (2020)) or arms inside the Pareto Set (multi-objective, see Drugan & Nowe (2013)) while satisfying some constraints (Amani et al., 2019; Li et al., 2024).

## 3. On the Complexity of Constrained PSI

In this section, we present some performance lower bounds for cPSI and e-cPSI. To do so, we first introduce some definitions to formalize the two objectives.

Given the polyhedron $P$ and a bandit instance $\nu \in \mathcal{D}^K$ with means $\boldsymbol{\mu} := (\mu_i)_{1 \leqslant i \leqslant K}$, we let $\mathrm{F}(\boldsymbol{\mu}) := \{i \mid \mu_i \in P\}$ be the set of arms that respect the linear constraints, called the *feasible set*.

For two arms $i, j$, arm $i$ is dominated by $j$ ($i \prec j$ or $\mu_i \prec \mu_j$) if $\mu_i^c \leqslant \mu_j^c$ holds for all $c \in [d]$. For $S \subset [K]$

and a parameter $\boldsymbol{\lambda} \in \mathcal{I}^K$, we let $\mathrm{Par}(S, \boldsymbol{\lambda})$ be the Pareto Set of $\{\lambda_i \mid i \in S\}$: the arms in $S$ that are not dominated by another arm in $S$ for the bandit $\boldsymbol{\lambda}$. With this notation, the Pareto Set of the feasible arm that we seek to identify is denoted by $O^*(\boldsymbol{\mu}) = \mathrm{Par}(F(\boldsymbol{\mu}), \boldsymbol{\mu})$ and we further let $\mathrm{SubOpt}(\boldsymbol{\mu}) := \{i \in [K] \mid \exists j \in \mathrm{F}(\boldsymbol{\mu}) \text{ such that } \mu_i \prec \mu_j\}$ be the set of arms that are dominated by a feasible arm. To simplify the notation, when $\boldsymbol{\mu}$ is clear from the context, we will sometimes write $F, O^*$ and $\mathrm{SubOpt}$.

### 3.1. Constrained PSI without explainability (cPSI)

In cPSI, an algorithm with stopping time $\tau$ simply outputs a recommendation $R_\tau = O_\tau$ which is a guess for $O^*$.

**Definition 3.1.** An algorithm is $\delta$-correct for cPSI on $\mathcal{D}^K$ if for any instance $\nu \in \mathcal{D}^K$, with means $\boldsymbol{\mu} \in \mathcal{I}^K$, $\mathbb{P}_\nu(\tau < \infty, O_\tau \neq O^*(\boldsymbol{\mu})) \leqslant \delta$.

We are interested in $\delta$-correct algorithms with a small sample complexity $\mathbb{E}_\nu[\tau_\delta]$. Given an instance $\nu$, to identify the (unique) correct answer, an algorithm must be able to differentiate $\nu$ with means $\boldsymbol{\mu}$ from any other instance $\nu'$ with means $\boldsymbol{\lambda}$ for which $O^*(\boldsymbol{\mu}) \neq O^*(\boldsymbol{\lambda})$. To state our lower bound for parametric models, we introduce $\mathrm{Alt}(O^*)$ to denote the set of instances of $\mathcal{I}^K$ with a different optimal set i.e., $\mathrm{Alt}(O^*) := \{\boldsymbol{\lambda} \in \mathcal{I}^K \mid O^*(\boldsymbol{\lambda}) \neq O^*\}$.

**Proposition 3.2.** *Let $\mathcal{D}$ be the class of multivariate normal with identity covariance and $\nu \in \mathcal{D}^K$ with means $\boldsymbol{\mu} \in (\mathbb{R}^d)^K$. For any $\delta$-correct algorithm for cPSI on $\mathcal{D}$ with stopping time $\tau$, it holds that*

$$\mathbb{E}_\nu[\tau] \geqslant T^*(\boldsymbol{\mu}) \log(1/(2.4\delta)) \text{ where,}$$

$$T^*(\boldsymbol{\mu})^{-1} := \max_{w \in \mathcal{S}_K} \inf_{\boldsymbol{\lambda} \in \mathrm{Alt}(O^*)} \sum_k \frac{1}{2} w_k \|\mu_k - \lambda_k\|_2^2. \quad (1)$$

The proof of this result follows a general recipe given by Garivier & Kaufmann (2016). Efficient algorithms designed to match similar bounds often rely on iterative saddle-point methods (Degenne et al., 2020) or online learning algorithms (Ménard, 2019; Wang et al., 2021). In any case, such algorithms require efficient solvers for the inner $\inf$ problem in (1), which is often a non-convex problem. In Appendix B, we propose an algorithm for cPSI of this flavor, called Game-cPSI (Algorithm 2), that matches the lower bound when $\delta$ is small. Moreover, we show that its computational cost is polynomial in the number of arms.

### 3.2. Constrained PSI with explainability (e-cPSI)

In the more demanding e-cPSI problem, an algorithm recommends a partition $R_\tau = (O_\tau, S_\tau, I_\tau)$ such that

$$i)\ O_\tau = O^*, \quad ii)\ S_\tau \subset \mathrm{SubOpt} \text{ and } I_\tau \subset \mathrm{F}^c$$

holds at stopping time $\tau$. Since some arms can be both infeasible and dominated by a feasible arm, such arms could be correctly classified either in $S_\tau$ or $I_\tau$. Therefore, multiple correct ways to partition $[K]$ may exist, so multiple correct $(S_\tau, I_\tau)$. Given the polyhedron $P$ we define the set of valid answers for $(S_\tau, I_\tau)$ as

$$\mathcal{M}(P, \boldsymbol{\mu}) := \{(S, I) \mid S \subset \mathrm{SubOpt}(\boldsymbol{\mu}), I \subset \mathrm{F}(\boldsymbol{\mu})^c,$$
$$S \cap I = \emptyset \text{ and } S \cup I = (O^*(\boldsymbol{\mu}))^c\}.$$

To ease the notation, we simply write $\mathcal{M}$ when $P$ and $\boldsymbol{\mu}$ are clear from the context. This is an example of a pure exploration problem with multiple correct answers, a framework studied by Degenne & Koolen (2019) in single objective bandits. With the notation above, the constrained PSI problem with explainability can be formalized as building a $\delta$-correct algorithm according to the following definition.

**Definition 3.3.** An algorithm is $\delta$-correct for e-cPSI on $\mathcal{D}^K$ if for any $\nu \in \mathcal{D}^K$ with means $\boldsymbol{\mu} \in \mathcal{I}^K$, it outputs a partition $(O_\tau, S_\tau, I_\tau)$ of $[K]$ s.t. $\mathbb{P}_\nu(\tau < \infty \text{ and } \neg(O_\tau = O^*, (S_\tau, I_\tau) \in \mathcal{M})) \leqslant \delta$.

Again, we are interested in $\delta$-correct algorithms with a small sample complexity $\mathbb{E}_\nu[\tau_\delta]$. To introduce the lower bound, given $(O, S, I)$ a partition of $[K]$ we introduce the set of instances for which $(O, S, I)$ is not a correct answer: $\mathrm{Alt}(S, I) := \{\boldsymbol{\lambda} \mid (S, I) \notin \mathcal{M}(P, \boldsymbol{\lambda})\}$.

**Proposition 3.4.** *Letting $\mathcal{D}$ be the class of multivariate normal with identity covariance and $\nu \in \mathcal{D}^K$ with means $\boldsymbol{\mu} \in \mathcal{I}^K$. For any $\delta$-correct algorithm for e-cPSI on $\mathcal{D}^K$ with stopping time $\tau$, it holds that*

$$\liminf_{\delta \to 0} \frac{\mathbb{E}_\nu[\tau]}{\log(1/\delta)} \geqslant T^*_{\mathcal{M}}(\boldsymbol{\mu}) := \min_{(S,I) \in \mathcal{M}} T(\boldsymbol{\mu}, S, I), \text{ where}$$

$$T(\boldsymbol{\mu}, S, I)^{-1} := \max_{w \in \mathcal{S}_K} \inf_{\boldsymbol{\lambda} \in \mathrm{Alt}(S,I)} \sum_k \frac{w_k}{2} \|\mu_k - \lambda_k\|_2^2. \quad (2)$$

This asymptotic lower bound derives from a game-theoretic technique used in Degenne & Koolen (2019). For $(S, I) \in \mathcal{M}$, $T(\boldsymbol{\mu}, S, I)$ quantifies the information-theoretic cost to identify $(S, I)$ as a valid answer. The lower bound states that the minimal sample complexity of any $\delta$-correct depends on the easiest-to-identify answer. Even if $\boldsymbol{\mu}$ was known, computing $T^*_{\mathcal{M}}(\boldsymbol{\mu})$ would be non-trivial as it involves solving the inner $\inf$ problem for each element of $\mathcal{M}$ which is of size $2^n$ for $n \geqslant 1$, and empty for $n = 0$, with $n := |\mathrm{SubOpt} \cap \mathrm{F}^c|$ satisfying $0 \leqslant n \leqslant K - 1$.

Degenne & Koolen (2019) proposed an optimal algorithm for similar single-objective bandit settings. However, their approach requires enumerating all possible answers at each step and solving the inner $\inf$ optimization (2) for each, making it computationally expensive. In our setting, this would require iterating over the power set of $\mathrm{SubOpt} \cap \mathrm{F}^c$, which can grow exponentially with $K$, making the approach impractical for large-scale problems.

Instead, we present in the next section a computationally efficient algorithm for e-cPSI based on upper and lower confidence bounds. We will show that its sample complexity is close to optimal in some instances, and assess its good empirical performance.

## 4. A Near-Optimal Algorithm for e-cPSI

Following the APE approach for (unconstrained) PSI from Kone et al. (2023), we base our algorithms on confidence regions on some quantities that allow to access whether an arm is dominated or not. For any pair of arms $i, j$, we introduce

$$\mathrm{M}(i,j) := \max_{c \leqslant d}[\mu_i^c - \mu_j^c] \,, \, \mathrm{m}(i,j) := \min_{c \leqslant d}[\mu_j^c - \mu_i^c] \,. \quad (3)$$

for which $\mathrm{M}(i,j) = -\mathrm{m}(j,i)$. Note that $\mathrm{M}(i,j) > 0$ if and only if $\mu_i$ is not dominated by $\mu_j$ ($\mu_i \nprec \mu_j$). We also introduce quantities expressing the hardness of accessing the feasibility of each arm $i$ (Katz-Samuels & Scott, 2018):

$$\eta_i := \begin{cases} \mathrm{dist}(\mu_i, P) & \text{if } \mu_i \notin P \,, \\ \mathrm{dist}(\mu_i, P^c) & \text{if } \mu_i \in P \,. \end{cases} \quad (4)$$

### 4.1. Constrained Adaptive Pareto Exploration

Our algorithm, called constrained Adaptive Pareto Exploration with explainability (e-cAPE) assumes that the arms are subGaussian and relies on confidence bounds to efficiently balance between feasibility detection and PSI. Its pseudo-code is given in Algorithm 1.

At any round $t$, we let $N_{t,i}$ denotes the number of pulls of arm $i$ up to time $t$, and $\hat{\mu}_{i,t}$ its empirical mean based on the $N_{t,i}$ samples collected. We let $\mathrm{M}(i,j;t), \mathrm{m}(i,j;t)$ and $\eta_i(t)$ denote the empirical versions of the quantities introduced in (3) and (4) (i.e., evaluated for the empirical means $(\hat{\mu}_{t,i})_{1 \leqslant i \leqslant K}$). $F_t := \{i \mid \hat{\mu}_{t,i} \in P\}$ is the empirical feasible set and $O_t := \mathrm{Par}(F_t, \hat{\mu}_t)$ the empirical Pareto Set of feasible arms with $\hat{\mu}_t := (\hat{\mu}_{t,i})_{1 \leqslant i \leqslant K}$.

We introduce some high-probability events

$$\mathcal{E}_t := \big\{ \forall i \in [K], \, \|\hat{\mu}_{t,i} - \mu_i\|_\infty \leqslant \beta_i(t, \delta) \text{ and}$$
$$\|\hat{\mu}_{t,i} - \mu_i\|_2 \leqslant U_i(t, \delta) \big\}, \quad \mathcal{E} := \bigcap_{t \geqslant 1} \mathcal{E}_t \,.$$

with confidence bonuses of the form

$$\beta_i(t, \delta) := \sqrt{\frac{2\sigma^2 f(t, \delta)}{N_{t,i}}} \text{ and } U_i(t, \delta) := \sqrt{\frac{2\sigma_u^2 g(t, \delta)}{N_{t,i}}} \,.$$

We assume that the functions $f, g$ (to be specified later) guarantee that $\mathbb{P}_\nu(\mathcal{E}) \geqslant 1 - \delta$ when $\nu$ is marginally $\sigma$-subGaussian and norm-$\sigma_u$-SubGaussian.

*Remark* 4.1. . When the arms are known to have independent marginals, we can set $\sigma_u = \sigma$. On the other extreme, when nothing is known on the covariance structure, we can always set $\sigma_u = \sqrt{d}\sigma$.

**Pareto Set Estimation** For each pair of arms $(i, j)$, we build time-uniform confidence intervals around the unknown quantities $\mathrm{M}(i, j)$ and $\mathrm{m}(i, j)$.

**Lemma 4.2.** *Under $\mathcal{E}_t$, for all $i, j$ it holds : (i) $|\mathrm{M}(i,j) - \mathrm{M}(i,j;t)| \leqslant \beta_i(t) + \beta_j(t)$ and $|\mathrm{m}(i,j) - \mathrm{m}(i,j;t)| \leqslant \beta_i(t) + \beta_j(t)$, (ii) for any $\mathcal{X} \subset \mathbb{R}^d$, $|\mathrm{dist}(\hat{\mu}_{t,i}, \mathcal{X}) - \mathrm{dist}(\mu_i, \mathcal{X})| \leqslant U_i(t)$.*

This result justifies the introduction of the confidence bounds $\mathrm{M}^\pm(i,j;t) = \mathrm{M}(i,j;t) \pm (\beta_i(t) + \beta_j(t))$ and similarly $\mathrm{m}^\pm(i,j;t) = \mathrm{m}(i,j;t) \pm (\beta_i(t) + \beta_j(t))$, such that $m(i,j) \in [\mathrm{m}^-(i,j;t), \mathrm{m}^+(i,j;t)]$ and $M(i,j) \in [\mathrm{M}^-(i,j;t), \mathrm{M}^+(i,j;t)]$ on the event $\mathcal{E}$. In particular on that event, $\mathrm{M}^-(i,j;t) > 0$ implies that $\mu_i \nprec \mu_j$ (arm $k$ is not dominated by $j$) and $\mathrm{M}^+(i,j;t) < 0$ implies that $\mu_i \prec \mu_j$ (arm $i$ is dominated by $j$).

**Feasibility Detection** For any arm $i$ introducing the quantity $\gamma_i(t) := \frac{1}{2\sigma_u^2} N_{t,i} \eta_i^2(t) - g(t, \delta)$, one can show using Lemma 4.2 that when $\gamma_i(t) > 0$, the vectors $\hat{\mu}_{i,t}$ and $\mu_i$ can either both belong to $P$ or to $P^c$, with high confidence. We let $G_t := \{i \in F_t^c \mid \gamma_i(t) < 0\}$ be the subset of $F_t^c$ of empirically infeasible arms that cannot be confidently ruled out as infeasible at time $t$.

**Stopping and Recommendation Rules** An algorithm for e-cPSI should stop as soon as any confidently valid answer has been identified. To introduce the stopping rule, we first define $Z_1^F(t) := \min_{i \in O_t} \gamma_i(t)$ and $Z_1^{\mathrm{PS}}(t) := \min_{i \in O_t} \min_{j \in O_t \setminus \{k\}} [\mathrm{M}^-(i,j;t)]$. We further define

$$Z_1(t) := \min(Z_1^F(t), Z_1^{\mathrm{PS}}(t)). \quad (5)$$

When $Z_1^F(t)$ is positive, arms in $O_t$ can be proved to be feasible, and having $Z_1^{\mathrm{PS}}(t)$ positive is sufficient to guarantee that arms in $O_t$ are not dominated by each other. Thus, when $Z_1(t) > 0$, arms in $O_t$ are all confidently feasible and non-dominated by each other. However, this alone is not sufficient to ensure that $O_t = O^*$. We also define $\xi_i(t) := \max_{j \in (F_t \cup G_t) \setminus \{k\}} \mathrm{m}^-(i,j;t)$ and note that $\xi_i(t) > 0$ implies that there exists $j \neq i \in (F_t \cup G_t)$ such that $\mu_i \prec \mu_j$. Therefore, $\mathbb{1}_{i \notin F_t}((\gamma_i(t) \vee \xi_i(t))) > 0$ implies that either arm $i$ is infeasible or it is dominated by an arm of $(F_t \cup G_t)$. We then introduce,

$$Z_2(t) := \min_{i \in [K] \setminus O_t} [\xi_i(t) \mathbb{1}_{i \in F_t} + (\gamma_i(t) \vee \xi_i(t)) \mathbb{1}_{i \notin F_t}], \quad (6)$$

for which a positive value guarantees that each arm outside of $O_t$ can be confidently classified as either infeasible or

dominated by another arm of $(F_t \cup G_t)$. We will prove that having both $Z_1(t) \geqslant 0$ and $Z_2(t) \geqslant 0$ for some $t$ is sufficient to prove that $O_t = O^*$ and to identify a correct answer $(S_t, I_t) \in \mathcal{M}$. Thus, we define the stopping time $\tau$ of our algorithm as

$$\tau := \inf\{t \geqslant 1 \mid Z_1(t) \geqslant 0 \quad \text{and} \quad Z_2(t) \geqslant 0\}. \quad (7)$$

At stopping, the algorithm recommends the partition $(O_\tau, S_\tau, I_\tau)$ of $[K]$ with $O_\tau = \mathrm{Par}(F_\tau, \hat{\boldsymbol{\mu}}_\tau)$, $S_\tau := (F_\tau \cup G_\tau) \backslash O_\tau$, a set of arms that will be shown to be dominated by some arms in $O_\tau$, and $I_\tau := G_\tau^c \cap F_\tau^c$, a set of arms deemed infeasible.

In Appendix C.1 we formally prove the correctness of the proposed stopping and recommendation rule.

**Lemma 4.3.** *On the event $\mathcal{E}$, if e-cAPE outputs $(O_\tau, S_\tau, I_\tau)$, we have $O_\tau = O^*$ and $(S_\tau, I_\tau) \in \mathcal{M}(P, \boldsymbol{\mu})$.*

The above result holds regardless of the sampling rule that is used, but the sampling rule is crucial to stop early.

**Sampling Rule** The challenge in designing an efficient sampling rule for constrained PSI lies in efficiently balancing the information about feasibility and Pareto dominance. We propose a sampling rule in the spirit of the top-two algorithms for best arm identification, of which LUCB (Kalyanakrishnan et al., 2012) can be viewed as the first instance. We define the leader $b_t$ greedily w.r.t the stopping rule, as follows:

$$b_t := \text{the minimizer in} \begin{cases} (6) & \text{if } Z_2(t) < 0 \\ (5) & \text{else.} \end{cases} \quad (8)$$

Intuitively, if the algorithm has not stopped, and $Z_2(t) < 0$, there is an arm of $O_t^c$ for which the evidence collected is not enough for it to be confidently classified as infeasible or dominated by an arm of $(F_t \cup G_t)$ so this arm should be pulled again. Similarly, if $Z_2(t) \geqslant 0$ and the algorithm has not stopped, then $Z_1(t) < 0$ so that the status of an arm in $O_t$ is uncertain and this arm should be pulled. Then we define the challenger of the arm $b_t$ as an arm that cannot yet be ruled out as infeasible and that is close to dominating $b_t$:

$$c_t := \underset{j \in (F_t \cup G_t) \backslash \{b_t\}}{\mathrm{argmin}} \mathrm{M}^-(b_t, j; t).$$

Pulling $b_t$ brings information about it being feasible or infeasible and further pulling $c_t$ brings information on whether $b_t$ is dominated by another arm of $(F_t \cup G_t)$ which as samples are collected will ultimately be the feasible set F. Sampling both $b_t$ and $c_t$ should increase the stopping statistic $Z(t) = \min(Z_1(t), Z_2(t))$.

### 4.2. Sample Complexity Guarantees

The analysis of cAPE features a new complexity quantity, that we first present. Additionally to $\eta_i$ which measures the

---

**Algorithm 1** e-cAPE

  Initialization : pull each arm once
  **for** $t = K + 1, K + 2, \ldots$ **do**
    Compute $F_t$ the empirical feasible set
    Compute $O_t := \{i \in F_t \mid \forall j \in F_t, \ \hat{\mu}_{t,i} \not\prec \hat{\mu}_{t,j}\}$
    **if** $Z_2(t) < 0$ **then**
      Get $b_t :=$ minimizer of (6)
    **else**
      Get $b_t :=$ minimizer of $\begin{cases} Z_1^F(t) & \text{if } Z_1^{\mathrm{PS}}(t) \geqslant Z_1^F(t) \\ Z_1^{\mathrm{PS}}(t) & \text{else.} \end{cases}$
    **end if**
    Get $c_t := \mathrm{argmin}_{j \in (F_t \cup G_t) \backslash \{b_t\}} \mathrm{M}^-(b_t, j; t)$
    **if** $\min(Z_1(t), Z_2(t)) \geqslant 0$ **then**
      **break** and **return** $(O_t, (F_t \cup G_t) \backslash O_t, F_t^c \cap G_t^c)$
    **end if**
    Pull arms $b_t$ and $c_t$
  **end for**

---

hardness of feasibility detection for arm $i$, it depends on some complexity measures from the PSI literature. Given $S \subset [K]$ a non-empty set, the complexity of identifying $\mathrm{Par}(S, \boldsymbol{\mu})$, the Pareto Set of $S$ in the unconstrained setting is governed by some "sub-optimality" gaps (Auer et al., 2016; Kone et al., 2023): for an arm $i \notin \mathrm{Par}(S, \boldsymbol{\mu})$,

$$\Delta_i(S) := \Delta_i^*(S) := \max_{j \in \mathrm{Par}(S, \boldsymbol{\mu})} \mathrm{m}(i, j), \quad (9)$$

which can be viewed as the smallest quantity that should be added component-wise to $\mu_i$ to make $i$ appear Pareto optimal w.r.t $\{\mu_j \mid j \in S \backslash \{i\}\}$. For an arm $i \in \mathrm{Par}(S, \boldsymbol{\mu})$,

$$\Delta_i(S) := \min(\delta_i^+(S), \delta_i^-(S)), \text{where} \quad (10)$$

$$\delta_i^+(S) := \min_{j \in \mathrm{Par}(S, \boldsymbol{\mu}) \backslash \{i\}} \min(\mathrm{M}(i, j), \mathrm{M}(j, i)), \text{and}$$

$$\delta_i^-(S) := \min_{j \in S \backslash \mathrm{Par}(S, \boldsymbol{\mu})} [(\mathrm{M}(j, i))_+ + \Delta_j],$$

with $\min_\emptyset = +\infty$. $\delta_i^+$ accounts for how much $i$ is close to dominating (or to be dominated by) another Pareto-optimal arm of $\mathrm{Par}(S, \boldsymbol{\mu})$ while $\delta_i^-$ quantifies the smallest "margin" from an optimal arm $i$ to the sub-optimal arms.

To identify $(S, I) \in \mathcal{M}$ as a correct answer, an algorithm should (i) identify the Pareto Set of $(O^* \cup S)$, (ii) certify that arms in $O^*$ are feasible and (iii) arms in $I$ are not feasible. We introduce below the quantity to measure the complexity of validating this candidate answer:

$$C(\nu, S, I) := \sum_{i \in O^*} \max\left( \frac{2\sigma^2}{\Delta_i^2(O^* \cup S)}, \frac{2\sigma_u^2}{\eta_i^2} \right)$$
$$+ \sum_{i \in S} \frac{2\sigma^2}{\Delta_i^2(O^* \cup S)} + \sum_{i \in I} \frac{2\sigma_u^2}{\eta_i^2}. \quad (11)$$

Interestingly, our upper bound scales with the complexity of the *easiest to verify* such answers:

$$C_{\mathcal{M}}^*(\nu) := \min_{(S,I) \in \mathcal{M}(P, \boldsymbol{\mu})} C(\nu, S, I) .$$

**Theorem 4.4.** *Let* $f(t, \delta) = \log(\frac{4k_1 K d t^\alpha}{\delta}), g(t, \delta) = 4\log(\frac{4k_1 K 5^d t^\alpha}{\delta})$ *with* $k_1 > 1 + \frac{1}{\alpha - 1}$ *and* $\mathcal{D}$ *the class of marginally $\sigma$-subGaussian and $\sigma_u$-norm-subGaussian distributions. Then for any $\alpha > 2$ and $\delta \in (0, 1)$, e-cAPE is $\delta$ correct on $\mathcal{D}^K$. Moreover, for all $\nu \in \mathcal{D}^K$ with means $\boldsymbol{\mu} \in \mathcal{I}^K$,*

$$\mathbb{E}_\nu[\tau] \leqslant 128 C_{\mathcal{M}}^*(\nu) \log\left(64 C_{\mathcal{M}}^*(\nu)\left(\frac{4k_1 K d}{\delta}\right)^{1/\alpha}\right) + 4H(\nu, \mathrm{F}^c) + \Lambda_\alpha ,$$

*where* $H(\nu, \mathrm{F}^c) := \sum_{a \in \mathrm{F}^c \cup O^*} \frac{32\sigma_u^2 \log(5^d/d)}{\eta_a^2}$ *and*

$$\Lambda_\alpha \leqslant \frac{2^{\alpha-1}}{4k_1} \sum_{T \geqslant 1} (\log(T)+1)\left(\frac{f(T, 5^{-d}d) + f(T, 1) + \frac{2}{e}}{T^{\alpha-1}}\right).$$

The proof of this result is given in Appendix C.2. It says that the sample complexity of cAPE essentially scales in $C_{\mathcal{M}}^*(\nu) \log(\frac{K C_{\mathcal{M}}^*(\nu)}{\delta})$ in the regime of small values of $\delta$. This is hard to compare in general to the lower bound of Proposition 3.4. However, we prove in Appendix E that the complexity of some particular instances has to scale with $C_{\mathcal{M}}^*(\nu)$, which makes our algorithm near-optimal in a worst-case sense.

**Theorem 4.5.** *There exists a class of multivariate Gaussian bandit instances* $\widetilde{\mathcal{D}}$ *such that any $\delta$-correct algorithm for e-cPSI satisfies, for all $\nu \in \mathcal{D}^K$ with means $\boldsymbol{\mu} \in \mathcal{I}^K$,*

$$\liminf_{\delta \to 0} \frac{\mathbb{E}_\nu[\tau_\delta]}{\log(1/\delta)} \geqslant \frac{C_{\mathcal{M}}^*(\nu)}{8}.$$

Moreover, since $(\mathrm{F} \backslash O^*, \mathrm{F}^c)$ is always a correct answer, we further observe that

$$C_{\mathcal{M}}^*(\nu) \leqslant C(\nu, \mathrm{F} \backslash O^*, \mathrm{F}^c)$$
$$= \sum_{i \in O^*} \max\left(\frac{2\sigma^2}{\Delta_i^2(\mathrm{F})}, \frac{2\sigma_u^2}{\eta_i^2}\right) + \sum_{i \in \mathrm{F} \backslash O^*} \frac{2\sigma^2}{\Delta_i^2(\mathrm{F})} + \sum_{i \notin \mathrm{F}} \frac{2\sigma_u^2}{\eta_i^2}$$
$$\leqslant \underbrace{\sum_{i \in \mathrm{F}} \frac{2\sigma^2}{\Delta_i^2(\mathrm{F})}}_{H_{\mathrm{PSI}}(\nu)} + \underbrace{\sum_{i \in [K]} \frac{2\sigma_u^2}{\eta_i^2}}_{H_F(\nu)} .$$

$H_F$ accounts for the complexity term for identifying the feasible set (Katz-Samuels & Scott, 2018) and $H_{\mathrm{PSI}}$ is the leading complexity of identifying the Pareto Set of the feasible arms (Auer et al., 2016; Kone et al., 2023). Thus,

the leading complexity term of e-cAPE is always smaller than that for the two-stage approach that first identifies the feasible set and then its Pareto Set. Compared to this baseline, e-cAPE does only pay for the "PSI cost" of arms in SubOpt $\cap$ F, and for arms in SubOpt $\cap$ F$^c$ it pays the smallest cost between infeasibility and Pareto dominance. We present in Figure 1 an example in which the complexity of cAPE can be arbitrarily smaller than that of the naive two-stage approach.

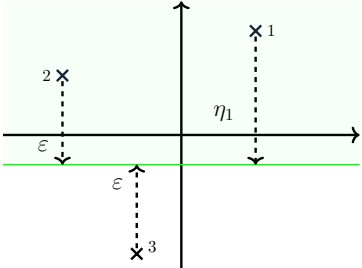

**Figure 1:** Example of an instance of constrained PSI. The complexity of a two-stage approach will scale with $1/\eta_1^2 + 2/\varepsilon^2 + \sum_{i=1}^2 1/\Delta_i(\{1,2\})^2$ while e-cAPE's complexity scales with $1/\eta_1^2 + \sum_{i=1}^3 1/\Delta_i(\{1,2,3\})^2$ when $\varepsilon \ll 1$.

Moreover, we remark that when the arms are very far from the boundaries of $P$, i.e., $\eta_i \gg 1$, e-cAPE matches the bounds of Auer et al. (2016) and Kone et al. (2023) for PSI on F. Similarly, when the PSI problem is easy (i.e., $\Delta_i \gg 1$) our guarantees are tight for feasibility detection.

**Improved Confidence Bounds** While we stated the guarantees of e-cAPE using specific confidence bounds of the form $\sqrt{\log(t^\alpha/\delta)/N_{t,i}}$ (with $\alpha > 2$), the correctness of e-cAPE can also be established under tighter bounds of the form $\sqrt{\frac{O(\log\log(N_{t,i}) + \log\frac{1}{\delta} + \log\log\frac{1}{\delta})}{N_{t,i}}}$, as prescribed by the law of the iterated logarithm (Jamieson et al., 2014; Kaufmann & Koolen, 2021). Although our techniques do not provide bounds on the expected stopping time under these tighter confidence intervals, we can, following the approach of Katz-Samuels & Scott (2019), show that an upper bound of the form

$$\tau_\delta \leqslant O\left(C_{\mathcal{M}}(\nu) \log\left(\frac{Kd}{\delta} \log(C_{\mathcal{M}}(\nu))\right)\right).$$

holds with probability at least $1 - \delta$.

As highlighted in the experimental section, we advocate for the use of such tighter confidence bonuses, as they yield better empirical performance, even though they remain conservative in terms of misidentification errors.

Furthermore, if each arm is assumed to be multivariate Gaussian with known covariance, applying the Hanson-Wright inequality for Gaussian vectors (Rudelson & Vershynin, 2013) allows us to achieve norm-2 concentration

bounds that depend on the trace and operator norm of the covariance matrix, rather than on exponential dependence in the dimension $d$, as is typically the case with covering arguments over the unit sphere (Katz-Samuels & Scott, 2018).

## 5. Experiments

We evaluate the performance of e-cAPE on two instances inspired by real-world data. We compare against the following baselines:

- *MD-APT+APE* (**A-A**): the two-stage algorithm that combines the optimal algorithms of Katz-Samuels & Scott (2018) and Kone et al. (2023) to first identify the feasible set then its Pareto Set

- *Uniform* (**U**): pull arms in a round-robin fashion until the stopping rule (7) is met

- *Racing algorithm* (**R-CP**): an adaptation of the elimination-based algorithm of Auer et al. (2016) for PSI to the e-cPSI problem, described in Appendix F.

In the experiments, (**A-A**), (**U**), (**R-P**) and e-cPSI are instantiated with the same confidence bonuses. Following the literature, we use slightly smaller thresholds than those licensed by theory and use the standard $\beta_i := \sqrt{2\sigma_i^2 \log(\log(t)/\delta)/N_{t,i}}$. Moreover, for the confidence bounds on M and m, instead of using individual confidence intervals, we use confidence intervals on pairs, as in the experiments of Auer et al. (2016) and Kone et al. (2023), and set $\beta_{i,j} = \sqrt{\beta_i^2 + \beta_j^2}$. We set the error parameter to $\delta = 0.1$ and we reported a negligible empirical error.

### 5.1. Datasets

We consider two datasets generated from real-word data extracted from clinical trials. Additional experiments on synthetic data are included in Appendix G.

**Secukinumab trial.** We use historical data from the phase 2 trial of Mark C et al. (2013), which evaluated the safety and efficacy of secukinumab for rheumatoid arthritis (also used by Katz-Samuels & Scott (2019) for constrained BAI). The trial involved 247 patients randomized to different doses (25mg, 75mg, 150mg, 300mg) and a placebo, with probabilities of efficacy and safety reported. Using average endpoints, we simulate a Bernoulli bandit, aiming to identify the Pareto Set of arms with safety above and non-toxicity above 40%. Figure 2 illustrates the feasible region and arm responses.

**CovBoost 19 trial** We simulate a constrained PSI bandit instance using data from Munro et al. (2021), which

reports average immune responses from a COVID-19 vaccine trial involving 20 strategies. Following Crepon et al. (2024), we focus on three indicators: neutralizing antibody titres (NT$_{50}$), immunoglobulin G (IgG), and wild-type cellular response. The unconstrained Pareto Set includes two strategies, and the feasible region is defined by arms with IgG response above 8.25 titre. Figure 3 shows the average responses and feasible region.

### 5.2. Results

We report the distribution of the sample complexity in Figure 4. For the experiment on covboost data, we excluded the *Uniform sampling* algorithm due to its excessively high sample complexity.

Our experiments show that e-cAPE performs well in both tasks, achieving a lower sample complexity than its competitors. In the dose-finding simulation, the 75mg arm is both near the feasibility boundary and suboptimal compared to the 150mg arm, which meets the constraints. By focusing on answers (or partitions) with minimal cost, e-cAPE can classify such arms as suboptimal, whereas (**A-A**), the two-stage algorithm, incurs an additional feasibility verification cost, which is high in this example. This finding is consistent with Figure 1, which highlights the sub-optimality of **A-A** when suboptimal arms lie near the boundary of the feasible region. Additionally, in Appendix F and Appendix G.3, we provide both theoretical and empirical evidence of the sub-optimality of the racing algorithm **R-CP**.

Appendix G.1 includes additional experiments on synthetic datasets, and details about the computational complexity of e-cAPE. These experiments confirm the robustness of e-cAPE to different scenarios (e.g. more complex polyhedra). We also compare in Appendix G.2 e-cAPE with our proposed optimal algorithm for the simpler cPSI objective, Game-cPSI (Algorithm 2). This preliminary experiment suggests that even for cPSI, e-cAPE may also be a good practical solution.

## 6. Conclusion and remarks

We introduced a novel multi-objective bandit pure exploration problem, where the goal of the leaner is to identify the Pareto Set while adhering to a set of linear feasibility constraints. We established the fundamental complexity of this problem by deriving an information-theoretic lower bound on the sample complexity required for any $\delta$-correct algorithm.

Our main contribution is e-cAPE, an algorithm that achieves near-optimal sample complexity in the explainable setting. We also propose an asymptotically optimal algorithm in the non-explainable case.

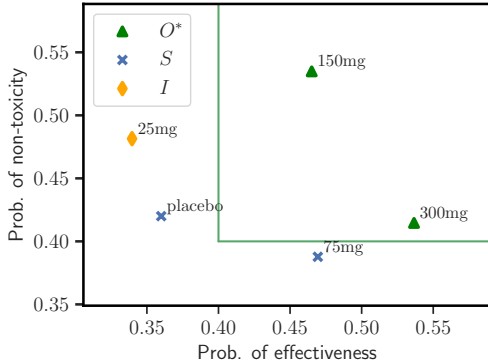

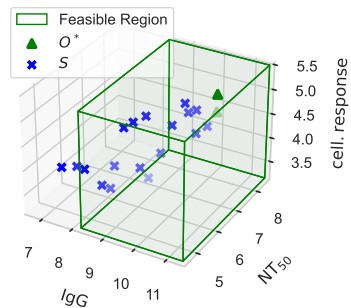

**Figure 2:** Average response for each dosage level. The feasible region is defined by an effectiveness of at least $40\%$ and non-toxicity

**Figure 3:** Average response of for each vaccine. The feasible region is defined by an `IgG` response larger than $8.25$ titre

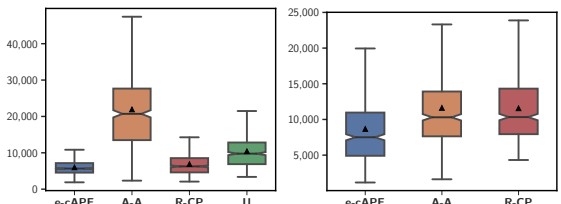

**Figure 4:** Empirical sample complexity averaged on 500 runs for the secukinumab trial (left) and the covboost trial (right).

Through our simulations from real-world and synthetic data, we demonstrate the superior empirical performance of e-cAPE, highlighting its efficiency for constrained Pareto Set Identification.

While our work ensures constraint satisfaction in the final recommendation, an exciting avenue for future research is to explore adaptive strategies that allow controlled constraint violations during the learning phase. This is particularly critical in applications like clinical trials, where carefully balancing exploration with safety constraints can lead to faster and more ethical decision-making.

## Acknowledgements

Cyrille Kone is funded by an Inria/Inserm PhD grant. This work has been partially supported by the French National Research Agency (ANR) in the framework of the PEPR IA FOUNDRY project (ANR-23-PEIA-0003).

## Impact Statement

This paper presents work whose goal is to advance the field of Machine Learning. There are many potential societal consequences of our work, none of which we feel must be specifically highlighted here.

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

# A. Outline and Notation

In Appendix B, we describe and analyze the complexity of an optimal algorithm for cPSI. The proof of the sample complexity of Algorithm 1 lies in Appendix C. Appendix D contains technical results used in the proofs. A gap-based lower bound for e-cPSI is proven in Appendix E. In Appendix F we analyze the limitations of a racing algorithm for e-cPSI. In Appendix G, additional experimental results are provided for more complex polyhedra and instances with a large number of arms. The experimental setup is also described in Appendix G for reproducibility.

**Table 1:** Notation for the setting.

| | |
|---|---|
| $\nu, \boldsymbol{\mu}$ | Bandit model and its matrix of mean parameters |
| $P$ | Polyhedron expressing the constraints : $\{x \in \mathbb{R}^d \mid \boldsymbol{A}x \leqslant b\}$ |
| $\partial P, \mathrm{int}(P)$ | Boundary and interior of the polyhedron $P$ |
| $F$ | Set of feasible arms $\{i \mid \mu_i \in P\}$ |
| $O^*$ | Pareto Set of the feasible arms |
| $\mathrm{Par}(S, \boldsymbol{\lambda})$ | Pareto Set of the vectors $\{\lambda_i \mid i \in S\}$ |
| $\mathrm{SubOpt}$ | Set of arms that are dominated by an arm of F |
| $B := \mathrm{SubOpt} \cap \mathrm{F^c}$ | Infeasible arms that are dominated by a feasible arm. |

We recall some commonly used notation: the set of integers $[K] := \{1, \cdots, K\}$, the complement $X^c$ and interior $\mathrm{int}(X)$ and the closure $\mathrm{cl}(X)$ of a set $X$, the indicator function $\mathbb{1}_X$ of an event, the probability $\mathbb{P}_\nu$ and the expectation $\mathbb{E}_\nu$ under distribution $\nu$. Landau's notation $o$, $O$, $\Omega$ and $\Theta$, the $(K-1)$-dimensional probability simplex $\mathcal{S}_K := \left\{ w \in \mathbb{R}_+^K \mid w \geqslant 0, \sum_{i \in [K]} w_i = 1 \right\}$. For $a, b \in \mathbb{R}$, $a \wedge b := \min(a, b)$ and $a \vee b := \max(a, b)$. We write $i \in F_t$ to when $\hat{\mu}_{t,i} \in P$ and $i \notin F_t$ otherwise.

Table 1 gathers problem-specific notation, and Table 2 groups notation for the algorithms.

**Table 2:** Notation for algorithms.

| | |
|---|---|
| $\tau$ | Stopping time of the algorithm |
| $\hat{\mu}_{t,k}$ | Sample mean of arm $k$ at time $t$ |
| $N_{t,k}$ | Number of pulls of arm $k$ up to time $t$ |
| $F_t$ | Empirical feasible set at time $t$ |
| $O_t$ | Empirical Pareto Set of feasible arms at time $t$ |

# B. An Asymptotically Optimal Algorithm for cPSI

In this section, we describe a $\delta$-correct algorithm tailored for the simpler cPSI problem (see Definition 3.1), in which the algorithm should identify $O^*(\boldsymbol{\mu})$ without the need to provide a classification for arms not in $O^*(\boldsymbol{\mu})$. We give an upper bound on its sample complexity, establishing its asymptotic optimality. We further analyze its computational cost and show that it is polynomial in the number of arms.

### B.1. A Game-Based algorithm for cPSI

We propose an algorithm for the constrained PSI problem. Our algorithm, Game-cPSI, is based on a game-theoretic interpretation of the lower bound. Indeed, as argued in Degenne & Koolen (2019), $T^{-1}(\boldsymbol{\mu})$ can be interpreted as the value of a two-player game between the algorithm which plays randomized strategies on the simplex and the nature selects the best alternative in $\mathrm{Alt}(O^*)$ to confuse the first player for identifying the optimal set $O^*$. The pseudo-code of Game-cPSI is described in Algorithm 2.

**Sampling Rule** We adopt the two-player iterative saddle-point solving scheme recently popularized by Degenne & Koolen (2019). We maintain a learner $\mathcal{L}_{\mathcal{S}_K}$ on the $(K-1)$ dimensional simplex $\mathcal{S}_K$. At each round, the recommen-

dation $O_t$ is computed from the empirical means and compute

$$\boldsymbol{\lambda}_t \coloneqq \operatorname*{argmin}_{\boldsymbol{\lambda} \in \operatorname{Alt}(O_t)} \sum_i w_{t,i} \| \hat{\mu}_{t,i} - \lambda_i \|_2^2 .$$

The mixed strategy $w_t$ of the sup player is computed from $\mathcal{L}_{\mathcal{S}_K}$ (for example using AdaHedge (De Rooij et al., 2014)) then the algorithm selects $k_t \sim w_t$ and pulls arm $k_t$. The learner $\mathcal{L}_{\mathcal{S}_K}$ is then updated with $g_t$, an optimistic value of the payoff, which we make explicit below.

**Stopping Rule** We use a GLRT (Generalized Likelihood Ratio Test) stopping rule. At each round, the GLRT statistic is computed, and the algorithm stops if this statistic exceeds a threshold $h(t, \delta)$, i.e., if

$$\inf_{\boldsymbol{\lambda} \in \operatorname{Alt}(O_t)} \sum_i \frac{N_{t,i}}{2} \| \hat{\mu}_{t,i} - \lambda_i \|_2^2 > h(t, \delta) ,$$

and the algorithm recommends $O_t$ at stopping. The threshold $h(t, \delta)$ is calibrated to ensure the correctness of the recommendation with high probability. In particular, the threshold is designed to ensure that for any $\delta \in (0, 1)$, the event

$$\Xi_\delta \coloneqq \left( \forall t \geqslant K, \sum_i \frac{N_{t,i}}{2} \| \hat{\mu}_{t,i} - \mu_i \|_2^2 \leqslant h(t, \delta) \right)$$

holds with probability at least $1 - \delta$. Introducing $c_{t,k} \coloneqq h(t^2, 1/t^3)/N_{t,k}$, we define

$$g_t(w) = \sum_i w_i (\| \hat{\mu}_{t,i} - \lambda_{t,i} \|_2 + \sqrt{c_{t,k}})^2 .$$

Using the property of the event $\Xi_\delta$ and triangular inequality, it is simple toe prove that this definition of $g_t$ guarantee optimism (i.e. $g_t(w) \geqslant \sum_i w_i \| \mu_i - \lambda_{t,i} \|_2^2$) with probability larger than $(1 - 1/t^2)$.

---

**Algorithm 2** Game-cPSI

---

*Initialize* : pull each arm once, $w_{K-1} = (1/K, \dots, 1/K)$
**for** $t = K, 2, \dots$ **do**
    Let $F_t \coloneqq \{ i \mid \boldsymbol{A} \hat{\mu}_{t,i} \leqslant b \}$ the empirical feasible set
    Let $O_t \coloneqq \{ i \in F_t \mid \forall j \in F_t \backslash \{i\}, \ \hat{\mu}_{t,i} \not\prec \hat{\mu}_{t,j} \}$
    **if** $\inf_{\boldsymbol{\lambda} \in \operatorname{Alt}(O_t)} \sum_i \frac{1}{2} N_{t,i} \| \hat{\mu}_{t,i} - \lambda_i \|_2^2 > h(t, \delta)$ **then**
        **break** and **return** $O_t$
    **end if**
    Get $w_t$ from the learner $\mathcal{L}_{\mathcal{S}_K}$
    Get $\boldsymbol{\lambda}_t \coloneqq \operatorname{argmin}_{\boldsymbol{\lambda} \in \operatorname{Alt}(O_t)} \sum_i w_{t,i} \| \hat{\mu}_{t,i} - \lambda_i \|_2^2$
    Update $\mathcal{L}_\Delta$ with gains $g_t(\cdot)$
    Get $k_t \sim w_t$
    Pull $k_t$ and collect observation $X_t$
    $t \leftarrow t + 1$
**end for**

---

### B.2. Theoretical guarantees

**Lemma B.1.** *(Kaufmann & Koolen, 2021) With the threshold $h(t, \delta) \coloneqq 2Kd \log(4 + \log(t/(Kd))) + Kd c_G \left( \frac{\log(1/\delta)}{Kd} \right)$, the GLRT stopping rule ensures that $\mathbb{P}_\nu(\tau < \infty, \widehat{O}_\tau \neq O^*(\boldsymbol{\mu})) \leqslant \delta$, for any $\nu \in \mathcal{D}^K$ with means $\boldsymbol{\mu} \in \mathcal{I}^K$ and $c_G(x) \approx x + \log(x)$.*

**Theorem B.2.** *Let $\mathcal{D}$ be the class of $d$-variate distributions with independent $1$-subGaussian marginals. For any $\nu \in \mathcal{D}^K$ with means $\boldsymbol{\mu} \in \mathcal{I}^K$, Game-cPSI satisfies*

$$\limsup_{\delta \to 0} \frac{\mathbb{E}_\nu[\tau_\delta]}{\log(1/\delta)} \leqslant T^*(\boldsymbol{\mu}).$$

The proof of this theorem follows from classical recipes developed in (Réda et al., 2021; Degenne et al., 2020). The specificity of our contribution is to come up with an efficient implementation method. Therefore, we focus on describing the procedure for computing the "best response" $\boldsymbol{\lambda}_t := \operatorname{argmin}_{\boldsymbol{\lambda} \in \operatorname{Alt}(O_t)} \sum_i w_{t,i} \|\hat{\mu}_{t,i} - \lambda_i\|_2^2$, which is the most critical part in the implementation of Game-cPSI.

## B.3. Computational cost

The main computational cost of Algorithm 2 is due to the computation of $\boldsymbol{\lambda}_t$. In a recent work, Crepon et al. (2024) proposed an algorithm to compute the best response $\boldsymbol{\lambda}_t$ in the case of unconstrained PSI (i.e., for $\boldsymbol{A} = \boldsymbol{0}, b \in \mathbb{R}_+^d$). The computational cost of their algorithm is $O\left((K(p+d) + d^3 p)\binom{p+d-1}{d-1}\right)$ where $p$ is the size of the Pareto Set. We will show that their algorithm can be adapted to the case of constrained PSI. We prove the following result, which allows us to decompose Alt into two simpler sets $\operatorname{Alt}^+(O)$ and $\operatorname{Alt}^-(O)$.

**Lemma B.3.** *It holds that*

$$\operatorname{Alt}(O) = \operatorname{Alt}^-(O) \cup \operatorname{Alt}^+(O)\,, where$$

$$
\begin{aligned}
\operatorname{Alt}^-(O) &:= \bigcup_{i,j \in O} \left(\{\boldsymbol{\lambda} \in \mathcal{I}^K \mid \boldsymbol{A}\lambda_i \not\leqslant b\} \cup \{\boldsymbol{\lambda} \mid \lambda_i \leqslant \lambda_j\}\right) \\
\operatorname{Alt}^+(O) &:= \bigcup_{i \notin O} \{\boldsymbol{\lambda} \in \mathcal{I}^K \mid \boldsymbol{A}\lambda_i \leqslant b \text{ and } \forall j \in O, \lambda_i \not\leqslant \lambda_j\}.
\end{aligned}
$$

*Proof.* We have

$$
\begin{aligned}
\boldsymbol{\lambda} \in \operatorname{Alt}(O) &\iff O^*(\boldsymbol{\lambda}) \neq O \\
&\iff (a): \exists i \in O \backslash O^*(\boldsymbol{\lambda}) \text{ or } (b): \exists i \notin O \text{ such that } i \in O^*(\boldsymbol{\lambda}) \\
&\iff (a): \exists (i,j) \in O^2, i \neq j \text{ s.t. } \lambda_i \notin P \text{ or } \lambda_i \prec \lambda_j \text{ or } (b): \exists i \notin O \text{ s.t. } \lambda_i \in P \text{ and } \forall j \in O, \lambda_i \not\prec \lambda_j
\end{aligned}
$$

To see the direct inclusion, let $\boldsymbol{\lambda} \in \operatorname{Alt}(O)$. If $O \not\subset \operatorname{F}(\boldsymbol{\lambda})$ then $(a)$ follows. Next, we assume $O \subset \operatorname{F}(\boldsymbol{\lambda})$.

As $O \neq \operatorname{Alt}(O)$, either $O \backslash O^*(\boldsymbol{\lambda}) \neq \emptyset$ or $O^*(\boldsymbol{\lambda}) \backslash O \neq \emptyset$. Assume there exists $i \in O$ such that $i \notin O^*(\boldsymbol{\lambda})$.

If $\lambda_i \notin P$, then, as in the case above, the inclusion follows. Assume $\lambda_i \in P$, i.e., $i$ is still a feasible arm in $\boldsymbol{\lambda}$. In this case, there exists $j \in O^*(\boldsymbol{\lambda})$ such that $\lambda_i \prec \lambda_j$, otherwise, we would have $i \in O^*(\boldsymbol{\lambda})$.

If $j \in O$, then the inclusion $(a)$ follows. If $j \notin O$, then, as $j \in O^*(\boldsymbol{\lambda})$, we have : $\lambda_j \in P$ and $\forall k \in \operatorname{F}(\boldsymbol{\lambda}), \lambda_j \not\prec \lambda_k$. In particular, as $O \subset \operatorname{F}(\boldsymbol{\lambda})$ it holds that for $k \in O$: $\lambda_j \not\prec \lambda_k$ and $j \in O^*(\boldsymbol{\lambda}) \subset \operatorname{F}(\boldsymbol{\lambda})$, so $(b)$ follows.

Now we assume there exists $i \in O^*(\boldsymbol{\lambda})$ such that $i \notin O$, as we have $i \in O^*(\boldsymbol{\lambda})$, it holds that $\lambda_i \in P$ and $\forall j \in \operatorname{F}(\boldsymbol{\lambda}), \lambda_i \not\prec \lambda_j$, in particular, as $O \subset \operatorname{F}(\boldsymbol{\lambda})$, we have $\forall j \in O, \lambda_i \not\prec \lambda_j$, then $(b)$ follows.

For the reverse inclusion, assume $(a)$ holds. Then, it follows directly that we cannot have $O = O^*(\boldsymbol{\lambda})$. Similarly, suppose $(b)$ holds and $O = O^*(\boldsymbol{\lambda})$. Then $\exists i \notin O = O^*(\boldsymbol{\lambda})$ such that $i \in \operatorname{F}(\boldsymbol{\lambda})$ and $\forall j \in O = O^*(\boldsymbol{\lambda}), \lambda_i \not\prec \lambda_j$, i.e., $i$ is feasible in the instance $\boldsymbol{\lambda}$, it does not belong to the optimal set $O^*(\boldsymbol{\lambda})$ and it is not dominated by any arm of $O^*(\boldsymbol{\lambda})$, which is impossible if $O = O^*(\boldsymbol{\lambda})$. Therefore, when $(a)$ or $(b)$ holds, we have $\boldsymbol{\lambda} \in \operatorname{Alt}(O)$. $\square$

**Lemma B.4.** *It holds that*

$$\inf_{\boldsymbol{\lambda} \in \operatorname{Alt}^-(O)} \sum_i \frac{1}{2} w_i \|\mu_i - \lambda_i\|_2^2 = \frac{1}{2} \min(\phi_1, \phi_2) \text{ where}$$

$$
\begin{aligned}
\phi_1 &:= \min_{i \in O} w_i \operatorname{dist}(\mu_i, P^c)^2, \\
\phi_2 &:= \min_{i,j \in O^2, i \neq j} \frac{w_i w_j}{w_i + w_j} \sum_{c \leqslant d} (\mu_i^c - \mu_j^c)_+^2,
\end{aligned}
$$

*and a minimizer can be computed in time $O(p^2 d + pdm)$ where $m$ is the number of constraints and $p = |O|$.*

*Proof.* We have

$$\text{Alt}^-(O) = \left( \bigcup_{i \in O} \Gamma_i \right) \bigcup \left( \bigcup_{\substack{i,j \in O^2 \\ i \neq j}} \Lambda_{i,j} \right),$$

with

$$\begin{aligned}
\Gamma_i &\coloneqq \{\boldsymbol{\lambda} \in \mathcal{I}^K \mid \lambda_i \notin P\} \\
\Lambda_{i,j} &\coloneqq \{\boldsymbol{\lambda} \in \mathcal{I}^K \mid \lambda_i \prec \lambda_j\}.
\end{aligned}$$

Therefore, letting $D_w(\boldsymbol{\lambda}) \coloneqq \sum_i \frac{1}{2} w_i \|\mu_i - \lambda_i\|_2^2$ we have

$$\inf_{\boldsymbol{\lambda} \in \text{Alt}^-(O)} D_w(\boldsymbol{\lambda}) = \left( \min_{i \in O} \inf_{\boldsymbol{\lambda} \in \Gamma_i} D_w(\boldsymbol{\lambda}) \right) \wedge \min_{i,j \in O^2, i \neq j} \inf_{\boldsymbol{\lambda} \in \Lambda_{i,j}} D_w(\boldsymbol{\lambda}) \right).$$

Next, we observe that

$$\inf_{\boldsymbol{\lambda} \in \Gamma_i} D_w(\boldsymbol{\lambda}) = \frac{1}{2} w_i \, \text{dist}(\mu_i, P^c)^2$$

and a minimizer of this quantity can be computed in $O(md)$. Indeed, since $\mu_i \in P$ (as $i \in O$), $\text{dist}(\mu_i, P^c) = \text{dist}(\mu_i, \partial P)$, and the latter can be computed in $O(md)$ for a polyhedron with matrix of constraints $\boldsymbol{A} \in \mathbb{R}^{m,d}$ (cf Lemma H.1 of Katz-Samuels & Scott (2018)). Finally Lemma 2 of Crepon et al. (2024) shows that the value of $\inf_{\boldsymbol{\lambda} \in \Lambda_{i,j}} D_w(\boldsymbol{\lambda})$ and its minimizer can be computed in time $O(p^2 d)$. $\qquad\square$

**Lemma B.5.** *There exists an algorithm that computes the value and a minimizer of $\inf_{\boldsymbol{\lambda} \in \text{Alt}^+(O)} \sum_i \frac{1}{2} w_i \|\mu_i - \lambda_i\|_2^2$ in polynomial time.*

*Proof.* In the instances of $\boldsymbol{\lambda} \in \text{Alt}^+(O)$ a novel arm is added to the Pareto Set of feasible arms. From its definition in Lemma B.3, we have

$$\text{Alt}^+(O) = \bigcup_{i \in O^c} \text{Alt}_i^+(O)$$

where

$$\text{Alt}_i^+(O) \coloneqq \{\boldsymbol{\lambda} \in \mathcal{I}^K \mid \lambda_i \in P \text{ and } \forall j \in O, \lambda_i \not\prec \lambda_j\}.$$

Then, note that to guarantee that arm $i$ is not dominated by $j$ while minimizing the transportation cost, it is sufficient to move the arm (or both arms) along one objective. Therefore, to minimize the transportation cost $D(w, \boldsymbol{\lambda})$, it is sufficient to move arm $i$ to a novel point $\lambda \in \mathbb{R}^d$ (satisfying the constraints) and move each arm $j \in O$ only along the coordinate with minimal cost to make $\lambda$ not dominated by the $(\lambda_j)_{j \in O}$. This amounts to solving the following optimization problem :

$$\inf_{\boldsymbol{\lambda} \in \text{Alt}_i^+(O)} D(w, \boldsymbol{\lambda}) = \inf_{\substack{\lambda \in \mathbb{R}^d \\ \boldsymbol{A}\lambda \leqslant b}} \frac{1}{2} w_i \|\mu_i - \lambda\|_2^2 + \sum_{j \in O} \frac{1}{2} w_j \min_{c \leqslant d} (\mu_j^c - \lambda^c)_+^2. \tag{12}$$

The rightmost problem is related to the optimization problem studied by Crepon et al. (2024), except that now we have additional linear constraints $\boldsymbol{A}\lambda \leqslant b$ due to the constrained PSI setting. However, we will show that their algorithm can still be adapted to efficiently solve (12). To see this, we let $\phi$ be a mapping from $O$ to $[d]$ and introduce $h_i^\phi : \mathbb{R}^d \to \mathbb{R}_+$, defined as

$$h_i^\phi(\lambda) \coloneqq \frac{1}{2} w_i \|\mu_i - \lambda\|_2^2 + \sum_{j \in O} \frac{1}{2} w_j (\mu_j^{\phi(j)} - \lambda^{\phi(j)})_+^2.$$

Then remark that the optimization problem in Equation 12 rewrites as

$$\min_{\phi \in [d]^p} \inf_{\substack{\lambda \in \mathbb{R}^d \\ \boldsymbol{A}\lambda \leqslant b}} h_i^\phi(\lambda), \tag{13}$$

where the notation $[d]^p$ denotes the set of mapping from $O$ to $[d]$ and $p = |O|$. However, as shown by Crepon et al. (2024), not all such maps are valid, and there is no need to optimize $h_i^\phi$ for an invalid map. The authors proposed an

efficient graph-based algorithm to enumerate all the valid maps. They showed that the number of valid maps is bounded by $\binom{p+d-1}{d-1}$. Now, it remains to minimize $h_i^\phi$ under the linear constraints for each valid map. Note that up to a reordering of the quantities, $(\mu_j^{\phi(j)})_{j \in O}$, $h_i^\phi$ is piecewise quadratic, and the problem can be decomposed into at most $p^d$ convex quadratic problems with constraints $\widetilde{A}\lambda \leqslant \widetilde{b}$ where $\widetilde{A} \in \mathbb{R}^{m+2d,d}$ and $\widetilde{b} \in \mathbb{R}^{m+2d}$. It is known that solving a convex QP can be done in polynomial time (Ye & Tse, 1989). Thus, when the number of arms is large and $d$ is small, the overall complexity of problem 13 is $O(p^{2d}s(m+d,d))$, where $s(m+2d,d)$ is the time complexity of a convex QP with $m+2d$ constraints in dimension $d$. $\qquad \square$

# C. Analysis of e-cAPE

In this section, we show the correctness of e-cAPE and upper bound its expected stopping time, showing a near-optimal sample complexity.

## C.1. Proof of the correctness

Before proving the correctness of e-cAPE, we introduce some notation. We recall that $P$ is the polyhedron of feasible arms, F is the set of indices of the feasible arms and

$$O^* := \{i \mid \mu_i \in P \text{ and } \forall j \text{ s.t } \mu_j \in P, \mu_i \nprec \mu_j\}$$

is the Pareto-optimal feasible set; $\mathrm{SubOpt} := \{i \in [K] \mid \exists j \in O^*\backslash\{i\} : \mu_i \prec \mu_j\}$ which can be rewritten as

$$\mathrm{SubOpt} := \{i \in [K] \mid \exists j \in \mathrm{F}\backslash\{i\} : \mu_i \prec \mu_j\}. \tag{14}$$

To see this, note that if $\mu_i \prec \mu_j$ and $j \in \mathrm{F}$, then, as $j \in \mathrm{F}$, either $j \in O^*$ or there exists $j' \in O^*$ such that $\mu_j \prec \mu_{j'}$ and so $\mu_i \prec \mu_{j'}$. Thus

$$\exists j \in \mathrm{F}\backslash\{i\} : \mu_j \prec \mu_i \implies \exists j \in O^*\backslash\{i\} : \mu_i \prec \mu_j \tag{15}$$

and the reverse inclusion is trivial, which justifies (14). Finally, it is simple to check that for the polyhedron $P$,

$$\mathrm{dist}(\mu, P^c) = \mathrm{dist}(\mu, \partial P) \quad \forall \mu \in P. \tag{16}$$

**Lemma C.1.** *On the event $\mathcal{E}$, when cAPE stops and returns a partition $(O_\tau, S_\tau, I_\tau)$, it is a valid partition.*

*Proof.* To show the correctness, we have to prove that the returned partition $(O_t, S_t, I_t)$ at $t = \tau$ satisfies $O_t = O^*, S_t \subset \mathrm{SubOpt}$ and $I_t \subset \mathrm{F}^c$. In this proof, we assume $\mathcal{E}$ holds and we let $t = \tau$.

**Step 1: Proving that $S_t \subset \mathrm{SubOpt}$.** Following (15), we have to show that

$$\forall i \in S_t, \exists j \in \mathrm{F} \text{ such that } \mu_i \prec \mu_j. \tag{17}$$

By design, $S_t := (F_t \cup G_t)\backslash O_t$ and at stopping

$$Z_2(t) := \min_{i \in [K]\backslash O_t}\left[\mathbb{1}_{i \in F_t}\left(\max_{j \in (F_t \cup G_t)\backslash\{i\}} \mathrm{m}^-(i,j;t)\right) + \mathbb{1}_{i \notin F_t}\left(\gamma_i(t) \vee \max_{j \in (F_t \cup G_t)\backslash\{i\}} \mathrm{m}^-(i,j;t)\right)\right]$$

satisfies $Z_2(t) \geqslant 0$ so that

$$\forall i \in S_t, \mathbb{1}_{i \in F_t}\left(\max_{j \in (F_t \cup G_t)\backslash\{i\}} \mathrm{m}^-(i,j;t)\right) + \mathbb{1}_{i \notin F_t}\left(\gamma_i(t) \vee \max_{j \in (F_t \cup G_t)\backslash\{i\}} \mathrm{m}^-(i,j;t)\right) \geqslant 0. \tag{18}$$

Remark that if $i \in S_t \cap G_t$, then $i \notin F_t$ and $\gamma_i(t) \leqslant 0$ so by (18),

$$\max_{j \in (F_t \cup G_t)\backslash\{i\}} \mathrm{m}^-(i,j;t) \geqslant 0,$$

which also holds if $i \in F_t$. Thus, it holds that

$$\forall i \in S_t, \max_{j \in (F_t \cup G_t) \setminus \{i\}} \mathrm{m}^-(i, j; t) \geqslant 0 \tag{19}$$

which on the event $\mathcal{E}$, using Lemma 4.2 and by the definition of $\mathrm{m}(i, j)$ yields

$$\forall i \in S_t, \ \exists j \in (F_t \cup G_t) \setminus \{i\} \text{ such that } \mu_i \prec \mu_j. \tag{20}$$

However, this does not directly imply (17) as the stopping rule only guarantees that $O_t \subset \mathrm{F}$ and at this point, some arms in $(F_t \cup G_t)$ could be infeasible (for the actual means). Thus, we shall prove that (20) holds for some $j \in O_t$. Remark that by their definition, we have $F_t \cup G_t = S_t \cup O_t$. Let

$$H_t := \mathrm{Par}(S_t, \boldsymbol{\mu}) = \{i \in S_t \mid \forall j \in S_t \setminus \{i\}, \mu_i \not\prec \mu_j\},$$

the Pareto Set of $S_t$ based on the true means.

As $H_t \subset S_t$, (20) applies and

$$\forall i \in H_t, \exists j \neq i, j \in (F_t \cup G_t) = (S_t \cup O_t) \text{ such that } \mu_i \prec \mu_j,$$

then, as $H_t$ is the Pareto Set of $S_t$, arms in $H_t$ cannot be dominated by another arm of $S_t$. Then, the equation above implies

$$\forall i \in H_t, \exists j \neq i, j \in O_t \text{ such that } \mu_i \prec \mu_j,$$

which as $O_t \subset \mathrm{F}$ yields

$$\forall i \in H_t, \exists j \in \mathrm{F} \setminus \{i\} \text{ such that } \mu_i \prec \mu_j. \tag{21}$$

Now, for $i \in S_t \setminus H_t$, there exists (by definition of $H_t$), $j \in H_t$ such that $\mu_i \prec \mu_j$ and by (21) there exists $j_2 \in \mathrm{F}$ such that $\mu_j \prec \mu_{j_2}$ so $\mu_i \prec \mu_{j_2}$. Put together, we have proved that

$$\forall i \in S_t, \ \exists j \in \mathrm{F} \setminus \{i\} \text{ such that } \mu_i \prec \mu_j,$$

hence $S_t \subset \mathrm{SubOpt}$.

**Step 2: Proving that $I_t \subset \mathrm{F}^c$.** The proof of this claim simply follows from the definition of $I_t$ and Lemma 4.2. Indeed, as $I_t := G_t^c \cap F_t^c \subset O_t^c$ and by definition of $G_t$, it holds that for all arm $i \in I_t, \gamma_i(t) \geqslant 0$. Then by definition of $\gamma_i$, the previous yields $\eta_i(t) \geqslant \sqrt{\frac{2\sigma_u^2 g(t, \delta)}{N_{t,i}}} := U(t, \delta)$ and since $i \in F_t^c$ we invoke Lemma 4.2, and

$$\begin{aligned} \mathrm{dist}(\mu_i, P) &> \mathrm{dist}(\hat{\mu}_{t,i}, P) - U_i(t, \delta) \\ &= \eta_i(t) - U_i(t, \delta) \geqslant 0, \end{aligned}$$

so $\mu_i \notin P$ that is $i \in \mathrm{F}^c$ which achieves to prove that $I_t \subset \mathrm{F}^c$.

**Step 3: Proving that $O_t = O^*$.** First, note that $(O_t, S_t, I_t)$ is a partition of $[K]$. Since we have proved that $S_t \subset \mathrm{SubOpt}$ and $I_t \subset \mathrm{F}^c$, and by definition $O^* \cap (\mathrm{SubOpt} \cup \mathrm{F}^c) = \emptyset$, thus it holds that $O^* \subset O_t$.

Then, noting that as $Z_1(t) \geqslant 0$, proceeding similarly to Step 2, we have $O_t \subset \mathrm{F}$, so at this step it holds that

$$O^* \subset O_t \text{ and } O_t \subset \mathrm{F}. \tag{22}$$

Moreover, from $Z_1(t) \geqslant 0$, we derive

$$\forall i \in O_t, \ \forall j \in O_t \setminus \{i\}, \ \mathrm{M}(i, j) \overset{\mathcal{E}}{>} \mathrm{M}^-(i, j; t) \geqslant 0$$

which from the definition of $\mathrm{M}(i, j)$ translates to

$$\forall i \in O_t, \ \forall j \in O_t \setminus \{i\}, \mu_i \not\prec \mu_j. \tag{23}$$

Therefore, as $O^* \subset O_t$ (22), we have

$$\forall i \in O_t, \ \forall j \in O^* \backslash \{i\}, \mu_i \not\prec \mu_j$$

which implies (using the transitivity of the Pareto dominance and the fact that any sub-optimal element in F is dominated by an element in $O^*$) that

$$\forall i \in O_t, \ \forall j \in \mathrm{F} \backslash \{i\}, \mu_i \not\prec \mu_j \ .$$

Moreover $O_t \subset \mathrm{F}$, so $O_t \subset O^*$ hence the conclusion follows as $O_t = O^*$.

Putting everything together, we have shown that the recommendation is a valid partition of $[K]$, which concludes the proof of the correctness of Algorithm 1 for e-cPSI (which also implies correctness of cPSI). □

*Remark* C.2. The proof of the correctness can be adapted to generic time-uniform confidence bounds on $\mathrm{M}^\pm, \mathrm{m}^\pm, \eta^\pm$ using an event $\mathcal{E} = \mathcal{E}_1 \cap \mathcal{E}_2$ where

$$\mathcal{E}_1 := \left(\forall t \geqslant 1, \forall (i,j) \in [K]^2, \mathrm{M}(i,j) \in [\mathrm{M}^-(i,j;t), \mathrm{M}^+(i,j;t)], \mathrm{m}(i,j) \in [\mathrm{m}^-(i,j;t), \mathrm{m}^+(i,j;t)]\right)$$

and

$$\mathcal{E}_2 := \left(\forall t \geqslant 1, \forall i \in [K], \eta_i \in [\eta_i^-(t), \eta_i^+(t)]\right)$$

such that $\mathcal{E}$ holds with probability at least $1 - \delta$.

## C.2. Stopping time and sample complexity

We upper bound the expected stopping time of algorithm 1. We recall that $P$ is fixed and known to the algorithm,

$$\mathcal{M}(P, \boldsymbol{\mu}) := \{(S, I) : S \subset \mathrm{SubOpt}, I \subset \mathrm{F}^c \text{ and } S \cap I = \emptyset, S \cup I = (O^*)^c\}$$

is the set of correct answers. The idea of the proof is to show that if the algorithm has not stopped after round $t$ and $\mathcal{E}_t$ holds, then at least one of $b_t, c_t$ has not been sufficiently pulled w.r.t. the cost of identifying any correct response of $\mathcal{M}$. More precisely, we introduce the sets

$$W_t^1(S) \ := \{i \in O^* : \Delta_i(O^* \cup S) \leqslant 4\beta_i(t, \delta) \text{ or } \eta_i \leqslant 2U_i(t, \delta)\}, \tag{24}$$
$$W_t^2(I) \ := \{i \in I : \eta_i \leqslant 2U_i(t, \delta)\}, \tag{25}$$
$$W_t^3(S) \ := \{i \in S : \Delta_i(O^* \cup S) \leqslant 4\beta_i(t, \delta)\}. \tag{26}$$

and finally, $W_t(S, I) = W_t^1(S) \cup W_t^2(I) \cup W_t^3(S)$. We may omit the dependency in $t$ when it is clear from the context. At round $t$, any arm that belongs to $W_t(S, I)$ is called under-explored w.r.t $(S, I)$. In the sequel, as $O^*$ is fixed and uniquely determined, we simplify notation: given $S \subset \mathrm{SubOpt}$, we write $\Delta_i(S)$ to denote $\Delta_i(S \cup O^*)$.

**Proposition C.3.** *If Algorithm 1 has not stopped at time $t$ and $\mathcal{E}_t$ holds then for all $(S, I) \in \mathcal{M}$, $\{b_t, c_t\} \cap W_t(S, I)$ is non-empty.*

### C.2.1. PROOF OF PROPOSITION C.3

In the remaining of this proof, we fix a correct answer $(S, I) \in \mathcal{M}$ and we show that if the algorithm has not stopped at time $t$ and $\mathcal{E}_t$ holds then one of $b_t, c_t$ is under-explored i.e $\{b_t, c_t\} \cap W_t(S, I) \neq \emptyset$. First, we address the case $(F_t \cup G_t) \backslash \{b_t\} = \emptyset$. Note that $(F_t \cup G_t) \backslash \{b_t\} = \emptyset$, implies that arms in $[K] \backslash \{b_t\}$ can all be classified as confidently infeasible at round $t$. Then,

$$O^* = \begin{cases} \{b_t\} & \text{if } b_t \in O^* \\ \emptyset & \text{else.} \end{cases}$$

In both cases, the gap of $b_t$ involves the quantity $\eta_{b_t}$. Assume $b_t \in \mathrm{F}$, then $b_t \in O^*$.

**Case 1.a**  If $b_t \in F_t$ then, as $G_t = \emptyset$ and $F_t \cup G_t = \{b_t\}$, we have

$$
\begin{aligned}
Z_2(t) & := \min_{i \in [K] \setminus O_t} \left[ \mathbb{1}_{i \in F_t} \left( \max_{j \in (F_t \cup G_t) \setminus \{i\}} \mathrm{m}^-(i,j;t) \right) + \mathbb{1}_{i \notin F_t} \left( \gamma_i(t) \vee \max_{j \in (F_t \cup G_t) \setminus \{i\}} \mathrm{m}^-(i,j;t) \right) \right] \\
& = \min_{i \neq b_t} \left[ \mathbb{1}_{i \in F_t} \left( \max_{j \in (F_t \cup G_t) \setminus \{i\}} \mathrm{m}^-(i,j;t) \right) + \mathbb{1}_{i \notin F_t} \left( \gamma_i(t) \vee \max_{j \in (F_t \cup G_t) \setminus \{i\}} \mathrm{m}^-(i,j;t) \right) \right] \\
& = \min_{i \neq b_t} \left[ \mathbb{1}_{i \notin F_t} \left( \gamma_i(t) \vee \max_{j \in (F_t \cup G_t) \setminus \{i\}} \mathrm{m}^-(i,j;t) \right) \right] \quad \text{(since } F_t = \{b_t\}\text{)} \\
& \geqslant 0
\end{aligned}
$$

which follows by definition of $G_t$, as $G_t = \emptyset$ and $F_t = \{b_t\}$. So if the algorithm has not stopped, $Z_1(t) < 0$. In this case, as $O_t$ is a singleton, we simply have

$$
Z_1(t) = \gamma_{b_t}(t) < 0
$$

(the reader can check the formula of $Z_1(t)$ as $F_t = \{b_t\}$, we recall $\min_\emptyset = \infty$). $\gamma_{b_t}(t) < 0$ then implies $\eta_i(t) \leqslant U_i(t,\delta)$.

Next, assume $b_t \in \mathrm{F}$, then $b_t \in O^*$ and

$$
\begin{aligned}
\eta_{b_t} = \mathrm{dist}(\mu_{b_t}, P^c) & \overset{\mathcal{E}_t}{\leqslant} \mathrm{dist}(\hat{\mu}_{t,b_t}, P^c) + U_{b_t}(t,\delta) \\
& = \eta_i(t) + U_{b_t}(t,\delta) \\
& \leqslant 2 U_{b_t}(t,\delta)
\end{aligned}
$$

where the first inequality follows from Lemma 4.2. Thus, if $b_t \in \mathrm{F}$ we have $b_t \in W_t^1(S)$. Now we assume $b_t \notin \mathrm{F}$. We observe that

$$
\begin{aligned}
\eta_{b_t} = \mathrm{dist}(\mu_{b_t}, P) & \overset{\mathcal{E}_t}{\leqslant} \mathrm{dist}(\hat{\mu}_{t,b_t}, P) + U_{b_t}(t,\delta) \\
& = 0 + U_{b_t}(t,\delta)
\end{aligned}
$$

which follows as by assumption $b_t \in F_t$. So we have $b_t \in W_t^2(I)$.

**Case 1.b**  If $b_t \in G_t$, then by definition $\gamma_{b_t}(t) = \mathrm{dist}(\hat{\mu}_{t,i}, P) - U_i(t,\delta) \leqslant 0$. By Lemma 4.2, either $b_t \notin \mathrm{F}$ and

$$
\begin{aligned}
\eta_{b_t} = \mathrm{dist}(\mu_{b_t}, P) & \overset{\mathcal{E}_t}{\leqslant} \mathrm{dist}(\hat{\mu}_{t,b_t}, P) + U_{b_t}(t,\delta) \\
& \leqslant 2 U_{b_t}(t,\delta)
\end{aligned}
$$

or $b_t \in \mathrm{F}$ (so $O^* = \{b_t\}$) and

$$
\begin{aligned}
\eta_{b_t} = \mathrm{dist}(\mu_{b_t}, P^c) & \overset{\mathcal{E}_t}{\leqslant} \mathrm{dist}(\hat{\mu}_{t,b_t}, P^c) + U_{b_t}(t,\delta) \\
& = 0 + U_{b_t}(t,\delta).
\end{aligned}
$$

Therefore, either $b_t \notin \mathrm{F}$ and $b_t \in W_t^2(I)$ or $b_t \in \mathrm{F} = \{b_t\} = O^*$ and $b_t \in W_t^1(S)$. This concludes the analysis for the case $(F_t \cup G_t) = \{b_t\}$.

In the sequel, we assume at round $t$, $(F_t \cup G_t) \setminus \{b_t\} \neq \emptyset$. For this proof, we treat different cases for $b_t, c_t$ in a series of lemmas summarized in the Table 3 which covers all the possible cases that could happen regarding $b_t, c_t$.

| Case | Reference |
|---|---|
| $b_t \in S$, and $c_t \in O^*$ | Lemma C.7 |
| $b_t \in O^*$ and $c_t \in O^*$ | Lemma C.6 |
| $b_t \in I$ or $c_t \in I$ | Lemma C.4 |
| $c_t \in S$ | Lemma C.5 |

**Table 3:** References to the exhaustive list of cases analyzed

The following lemmas are proved under the condition that cAPE has not stopped and $\mathcal{E}_t$ holds.

**Lemma C.4.** *If the cAPE has not stopped and $b_t \in I$ or $c_t \in I$ then one of them satisfies $\eta_i \leqslant 2U_i(t,\delta)$, i.e. $W_t^2(I) \cap \{b_t, c_t\} \neq \emptyset$.*

*Proof.* By design of $b_t$ it holds that

$$\text{dist}(\hat{\mu}_{t,b_t}, P) \leqslant U_i(t,\delta). \tag{27}$$

To see this, observe that if $Z_2(t) < 0$, then, as in this case

$$b_t := \underset{i \in O_t}{\text{argmin}} \left[ \gamma_i(t) \wedge \min_{j \in O_t} \text{M}^-(i,j;t) \right]$$

and $O_t \subset F_t$ we have $\hat{\mu}_{t,b_t} \in P$, so (27) trivially holds. If otherwise $Z_2(t) > 0$ and $Z_1(t) \leqslant 0$ then, recalling that in this case

$$b_t = \underset{i \in [K] \setminus O_t}{\text{argmin}} \left[ \mathbb{1}_{i \in F_t} \left( \max_{j \in (F_t \cup G_t) \setminus \{i\}} \text{m}^-(i,j;t) \right) + \mathbb{1}_{i \notin F_t} \left( \gamma_i(t) \vee \max_{j \in (F_t \cup G_t) \setminus \{i\}} \text{m}^-(i,j;t) \right) \right]$$

it holds that either i) $b_t \in F_t$ so $\hat{\mu}_{t,b_t} \in P$ and (27) holds or ii) $b_t \notin F_t$ and $\gamma_{b_t}(t) \leqslant 0$ (otherwise $Z_2(t)$ would be non-negative). which by definition of $\gamma_{b_t}(t)$ yields Equation (27).

Next, if $b_t \in I$, then, as the event $\mathcal{E}_t$ holds, Lemma 4.2 yields

$$\eta_{b_t} := \text{dist}(\mu_{b_t}, P) \leqslant \text{dist}(\hat{\mu}_{t,b_t}, P) + U_{b_t}(t,\delta)$$

and combining with (27) yields $\eta_{b_t} \leqslant 2U_{b_t}(t)$. Assume instead $c_t \in I$. In this case, as $c_t \in F_t \cup G_t$ we either have $\text{dist}(\hat{\mu}_{t,c_t}, P) = 0$ (when $c_t \in F_t$) or $\gamma_{b_t}(t) \leqslant 0$ (when $c_t \in G_t \subset F_t^c$) i.e $\text{dist}(\hat{\mu}_{t,b_t}, P) \leqslant U_{b_t}(t,\delta)$ so that (27) holds. Thus, the proof follows as in the previous case, leading to the conclusion that

$$\eta_{c_t} \leqslant 2U_{c_t}(t,\delta),$$

and so $W_t^2(I) \cap \{b_t, c_t\} \neq \emptyset$. $\qquad \square$

**Lemma C.5.** *If cAPE has not stopped and $c_t \in S$ then one of $b_t, c_t$ is under-explored.*

*Proof.* If $c_t \in S \subset \text{SubOpt}$, then there exists $c_t^* \in O^*$ such that

$$\Delta_{c_t}(S) = \text{m}(c_t, c_t^*).$$

As $c_t^* \in \text{F}$, it is easy to see that, on the event $\mathcal{E}_t$, $c_t^*$ can not be declared as confidently infeasible at time $t$, so $c_t^* \in F_t \cup G_t$. If $b_t \neq c_t^*$ then by definition of

$$c_t := \underset{j \in (F_t \cup G_t) \setminus \{b_t\}}{\text{argmax}} \text{m}^+(b_t, j; t)$$

it holds that $\text{m}^+(b_t, c_t; t) \geqslant \text{m}^+(b_t, c_t^*; t)$, which by definition of $\text{m}^+(i,j;t)$ yields

$$\exists c \in [d] : \hat{\mu}_{t,c_t}^c - \hat{\mu}_{t,b_t}^c + \beta_{b_t}(t,\delta) + \beta_{c_t}(t,\delta) \geqslant \hat{\mu}_{t,c_t^*}^c - \hat{\mu}_{t,b_t} + \beta_{b_t}(t,\delta) + \beta_{c_t^*}(t,\delta)$$

thus when $\mathcal{E}_t$ holds this implies that

$$\exists c \in [d] : \mu_{c_t^*}^c - \mu_{c_t}^c \leqslant 2\beta_{c_t}(t,\delta)$$

which in turn, implies

$$\Delta_{c_t}(S, I) \leqslant 2\beta_{c_t}(t,\delta). \tag{28}$$

If otherwise $b_t = c_t^* \in O^*$ then as the algorithm has not stopped, either $Z_2(t) < 0$ or $Z_1(t) < 0$ holds. We analyze both cases below.

**Case 1:** $Z_2(t) < 0$. In this case, as by design $b_t \in O_t^c$, either

a) $b_t \in F_t$ and there exists $j \in O_t$ such that $\hat{\mu}_{t,b_t} \prec \hat{\mu}_{t,j}$, that is $b_t$ is empirically feasible sub-optimal or

b) $b_t \notin F_t$ holds.

In the case $a$), for the arm $j \in O_t$, as $\hat{\mu}_{t,b_t} \prec \hat{\mu}_{t,j}$, it holds that $\mathrm{M}(b_t, j; t) \leqslant 0$ so $\mathrm{M}^-(b_t, j; t) \leqslant 0$. So, as $j \in (F_t \cup G_t)$ and by definition of $c_t$ it holds that $\mathrm{M}^-(b_t, c_t; t) \leqslant 0$ i.e

$$\mathrm{M}(b_t, c_t; t) \leqslant \beta_{b_t}(t, \delta) + \beta_{c_t}(t, \delta), \tag{29}$$

then, note that as $b_t = c_t^*$, $\Delta_{b_t}(S) \leqslant \Delta_{c_t}(S)$ (this follows from (10)) so

$$
\begin{aligned}
\max(\Delta_{b_t}(S), \Delta_{c_t}(S)) \quad &\leqslant \quad \Delta_{c_t}(S) := \mathrm{m}(c_t, c_t^*) = \mathrm{m}(c_t, b_t) \\
&\leqslant \quad \mathrm{M}(b_t, c_t) \\
&\overset{(i)}{\leqslant} \quad \mathrm{M}(b_t, c_t; t) + \beta_{b_t}(t, \delta) + \beta_{c_t}(t, \delta) \\
&\overset{(29)}{\leqslant} \quad 2(\beta_{b_t}(t, \delta) + \beta_{c_t}(t, \delta))
\end{aligned}
$$

where $(i)$ follows from Lemma 4.2. Thus, for $a_t$, the least explored among $b_t, c_t$, it holds that

$$\Delta_{a_t}(S) \leqslant 4\beta_{a_t}(t, \delta),$$

which implies that $a_t \in W_t(S, I)$ in the case $a$). In the sub-case $b$), we have $b_t \notin F_t$, as $b_t = c_t^* \in O^*$, we have

$$
\begin{aligned}
\eta_{b_t} \quad &= \quad \mathrm{dist}(\mu_{b_t}, P^c) \\
&\leqslant \quad \mathrm{dist}(\hat{\mu}_{t,b_t}, P^c) + U_{b_t}(t, \delta) \quad \text{(by Lemma 4.2 on } \mathcal{E}_t) \\
&\leqslant \quad U_{b_t}(t, \delta),
\end{aligned}
$$

$b_t \in W_t(S, I)$ and this concludes the analysis of Case 1.

**Case 2:** $Z_1(t) < 0$ **and** $Z_2(t) \geqslant 0$**.** In this case, by design, $b_t \in O_t$. As

$$Z_1(t) := \min_{i \in O_t} \left[ \gamma_i(t) \wedge \min_{j \in O_t \setminus \{i\}} [\mathrm{M}^-(i, j; t)] \right]$$

is negative, and $b_t$ is its minimizer, either $a$): there exists $j \in O_t$ such that $\mathrm{M}^-(b_t, j; t) \leqslant 0$, which further yields

$$\mathrm{M}^-(b_t, c_t, t) \leqslant 0$$

or b): $\gamma_{b_t}(t) < 0$. In the case $a$), proceeding similarly to Case 1.$a$) will lead to the conclusion that $a_t$, the least explored arm among $b_t, c_t$ belongs to $W(S, I)$. The latter sub-case $b$) leads to $\mathrm{dist}(\hat{\mu}_{t,b_t}, P^c) \leqslant U_{b_t}(t, \delta)$ so, as $b_t = c_t^* \in O^*$, again we have

$$
\begin{aligned}
\eta_{b_t} \quad &= \quad \mathrm{dist}(\mu_{b_t}, P^c) \\
&\leqslant \quad \mathrm{dist}(\hat{\mu}_{t,b_t}, P^c) + U_{b_t}(t, \delta) \quad \text{(by Lemma 4.2 on } \mathcal{E}_t) \\
&\leqslant \quad 2U_{b_t}(t, \delta),
\end{aligned}
$$

that is $b_t \in W_t(S, I)$. Therefore, we conclude that one of $b_t$ or $c_t$ is under-explored. $\qquad \square$

**Lemma C.6.** *If $b_t \in O^*$ and $c_t \in O^*$, then one of them is under-explored.*

*Proof.* Note that in this case, by definition of the gaps (cf (10) and (4.2)), as $b_t, c_t \in O^*$,

$$\Delta_{b_t}(S) \leqslant \mathrm{M}(b_t, c_t) \text{ and } \Delta_{c_t}(S) \leqslant \mathrm{M}(b_t, c_t). \tag{30}$$

We analyze first the case $Z_1(t) < 0$, $Z_2(t) \geqslant 0$. By design,

$$b_t := \underset{i \in O_t}{\mathrm{argmin}} \left[ \gamma_i(t) \wedge \min_{j \in O_t \setminus \{i\}} \mathrm{M}^-(i, j; t) \right],$$

so, as $Z_1(t) < 0$, either $b_t$ is not confidently feasible (i.e $\gamma_{b_t}(t) \leqslant 0$ in which case the proof is similar to Lemma C.4) or there exists $j \in O_t \setminus \{b_t\}$ such that $\mathrm{M}^-(b_t, j; t) \leqslant 0$. Which as $j \in F_t$ (since $O_t \subset F_t$) and by design

$$c_t := \underset{j \in (F_t \cup G_t) \setminus \{b_t\}}{\mathrm{argmin}} [\mathrm{M}^-(b_t, j; t)],$$

it follows that
$$\mathrm{M}^-(b_t, c_t; t) \leqslant 0. \tag{31}$$

Then, using concentration properties of $\mathrm{M}(i,j;t)$ (cf Lemma 4.2) and from (30), it follows

$$
\begin{aligned}
\max(\Delta_{b_t}, \Delta_{c_t}) &\leqslant \mathrm{M}^+(b_t, c_t, t) \\
&\leqslant \mathrm{M}(b_t, c_t; t) + \beta_{b_t}(t, \delta) + \beta_{c_t}(t, \delta) \\
&\leqslant 2(\beta_{b_t}(t, \delta) + \beta_{c_t}(t, \delta))
\end{aligned}
$$

where the last inequality follows from $\mathrm{M}^-(b_t, c_t; t) \leqslant 0$. For $a_t$ the least-explored among $b_t, c_t$, the latter inequality implies

$$\Delta_{a_t}(S) \leqslant 4\beta_{a_t}(t, \delta),$$

that is $a_t \in W_t(S, I)$, which concludes our proof in the case $Z_1(t) < 0, Z_2(t) \geqslant 0$.

In the case $Z_2(t) < 0$, we recall that

$$b_t = \operatorname*{argmin}_{i \in [K] \setminus O_t} \left[ \mathbb{1}_{i \in F_t} \left( \max_{j \in (F_t \cup G_t) \setminus \{i\}} \mathrm{m}^-(i,j;t) \right) + \mathbb{1}_{i \notin F_t} \left( \gamma_i(t) \vee \max_{j \in (F_t \cup G_t) \setminus \{i\}} \mathrm{m}^-(i,j;t) \right) \right].$$

Assume $b_t \notin F_t$ (i.e. $\hat{\mu}_{t,b_t} \notin P$). Then by design, since $Z_2(t) < 0$ we have $\gamma_{b_t}(t) < 0$ and using concentration properties of Lemma 4.2, and as $b_t \in O^*$,

$$
\begin{aligned}
\eta_{b_t} &= \mathrm{dist}(\mu_{b_t}, P^c) \\
&\leqslant \mathrm{dist}(\hat{\mu}_{t,b_t}, P^c) + U_{b_t}(t, \delta), \quad \text{Lemma 4.2 on } \mathcal{E}_t \\
&\leqslant U_{b_t}(t, \delta),
\end{aligned}
$$

which follows as $\hat{\mu}_{t,b_t} \in P^c$. Thus $b_t \in W_t(S, I)$.

If $b_t \in F_t$ (i.e. $\hat{\mu}_{t,b_t} \in P$), as by design $b_t \in O_t^c$, then $b_t \in O_t^c \cap F_t$ so $b_t$ is empirically feasible sub-optimal and

$$\exists j \in O_t \text{ such that } \hat{\mu}_{t,b_t} \prec \hat{\mu}_{t,j}$$

that is there exists $\exists j \in F_t \setminus \{b_t\}$ such that $\mathrm{M}(b_t, j; t) \leqslant 0$, which as $\mathrm{M}^-(b_t, j; t) \leqslant \mathrm{M}(b_t, j; t)$ for all $j$ yields

$$\exists j \in F_t \setminus \{b_t\} \text{ such that } \mathrm{M}^-(b_t, j; t). \tag{32}$$

Therefore, by design of $c_t$,
$$\mathrm{M}^-(b_t, c_t; t) \leqslant 0 \tag{33}$$

so that we have with (30),

$$
\begin{aligned}
\max(\Delta_{b_t}(S), \Delta_{c_t}(S)) &\leqslant \mathrm{M}^+(b_t, c_t, t) \\
&\overset{\mathcal{E}_t}{\leqslant} \mathrm{M}(b_t, c_t; t) + \beta_{b_t}(t, \delta) + \beta_{c_t}(t, \delta) \\
&\overset{(33)}{\leqslant} 2(\beta_{b_t}(t, \delta) + \beta_{c_t}(t, \delta))
\end{aligned}
$$

so, for $a_t$, the least explored arm among $b_t, c_t$

$$\Delta_{a_t}(S) \leqslant 4\beta_{a_t}(t, \delta),$$

that is $a_t \in W_t(S, I)$. $\qquad \square$

**Lemma C.7.** *If $b_t \in S$ and $c_t \in O^*$ then either $b_t$ or $c_t$ is under explored.*

*Proof.* If $Z_2(t) \leqslant 0$ then by design, as

$$b_t = \operatorname*{argmin}_{i \in [K] \setminus O_t} \left[ \mathbb{1}_{i \in F_t} \left( \max_{j \in (F_t \cup G_t) \setminus \{i\}} \mathrm{m}^-(i,j;t) \right) + \mathbb{1}_{i \notin F_t} \left( \gamma_i(t) \vee \max_{j \in (F_t \cup G_t) \setminus \{i\}} \mathrm{m}^-(i,j;t) \right) \right],$$

and

$$Z_2(t) := \min_{i \in [K] \backslash O_t} \left[ \mathbb{1}_{i \in F_t} \left( \max_{j \in (F_t \cup G_t) \backslash \{i\}} \mathrm{m}^-(i,j;t) \right) + \mathbb{1}_{i \notin F_t} \left( \gamma_i(t) \vee \max_{j \in (F_t \cup G_t) \backslash \{i\}} \mathrm{m}^-(i,j;t) \right) \right]$$

plus $c_t \in (F_t \cup G_t) \backslash \{b_t\}$, we have $\mathrm{m}^-(b_t, c_t; t) \leqslant 0$. We invoke concentration properties on $\mathcal{E}_t$ to have

$$\Delta_{b_t}(S) = \mathrm{m}(b_t, b_t^*) \leqslant \mathrm{m}^+(b_t, b_t^*; t)$$

where $b_t^* := \operatorname{argmax}_{j \in \mathrm{F}} \mathrm{m}(b_t, j)$ and note that on the event $\mathcal{E}_t$, it holds that $b_t^* \in (F_t \cup G_t)$ so by definition of $c_t$, $\Delta_{b_t} \leqslant \mathrm{m}^+(b_t, c_t; t)$ and replacing by its expression, we further get

$$\Delta_{b_t}(S) \leqslant 2(\beta_{b_t}(t, \delta) + \beta_{c_t}(t, \delta)) .$$

On the other side, observe that by definition of the gaps and as $b_t \in S$,

$$\Delta_{c_t}(S) \leqslant \mathrm{M}(b_t, c_t)_+ + \Delta_{b_t}(S) \tag{34}$$

which then yields

$$\begin{aligned} \Delta_{c_t}(S) &\leqslant (-\mathrm{m}(b_t, c_t; t) + \beta_{b_t}(t, \delta) + \beta_{c_t}(t, \delta))_+ + \mathrm{m}(b_t, c_t; t) + \beta_{b_t}(t, \delta) + \beta_{c_t}(t, \delta) \\ &\leqslant \max \left( \mathrm{m}(b_t, c_t; t) + \beta_{b_t}(t, \delta) + \beta_{c_t}(t, \delta), 2(\beta_{b_t}(t, \delta) + \beta_{c_t}(t, \delta)) \right) \end{aligned}$$

thus

$$\Delta_{c_t}(S) \leqslant 2(\beta_{b_t}(t, \delta) + \beta_{c_t}(t, \delta)) \text{ and } \Delta_{b_t}(S) \leqslant 2(\beta_{b_t}(t, \delta) + \beta_{c_t}(t, \delta)),$$

so we conclude in the case $Z_2(t) \leqslant 0$.

Now, we assume $Z_2(t) > 0$; then $Z_1(t) < 0$ (otherwise, the algorithm would have stopped). We still have

$$\Delta_{b_t}(S) = \mathrm{m}(b_t, b_t^*) \leqslant \mathrm{m}^+(b_t, b_t^*; t) \leqslant \mathrm{m}^+(b_t, c_t; t) ,$$

then, the idea is to show that $\mathrm{m}^-(b_t, c_t, t) \leqslant 0$. We will prove that for any $i \in (F_t \cup G_t), \mathrm{m}^-(b_t, i, t) \leqslant 0$. To see this, first, note that, as $b_t \in O_t$, for any $i \in F_t, \hat{\mu}_{t,b_t} \not\prec \hat{\mu}_{t,i}$, thus for any $i \in F_t, \mathrm{m}(b_t, i, t) \leqslant 0$.

Now let $i \in G_t$. By Lemma C.8, there exists $j \in O_t$ such that $\hat{\mu}_{t,i} \prec \hat{\mu}_{t,j}$.

Having $\hat{\mu}_{t,b_t} \prec \hat{\mu}_{t,i}$ would yield either $\hat{\mu}_{t,b_t} \prec \hat{\mu}_{t,i} \prec \hat{\mu}_{t,b_t}$ (if $j = b_t$) or by transitivity

$$\hat{\mu}_{t,b_t} \prec \hat{\mu}_{t,j} ,$$

with $j \neq b_t$, which is not possible for $b_t, j \in O_t$. Therefore, for any $i \in F_t, \mathrm{m}(b_t, i; t) \leqslant 0$, so combined with the previous display,

$$\forall i \in (F_t \cup G_t), \mathrm{m}(b_t, i; t) \leqslant 0 .$$

In particular, $\mathrm{m}(b_t, c_t; t) \leqslant 0$. Moreover, recalling that on $\mathcal{E}_t$,

$$\Delta_{b_t}(S) \leqslant \mathrm{m}^+(b_t, c_t; t) ,$$

we have $\Delta_{b_t}(S) \leqslant \beta_{b_t}(t, \delta) + \beta_{c_t}(t, \delta)$. On the other side, as in (34),

$$\Delta_{c_t}(S) \leqslant \mathrm{M}(b_t, c_t)_+ + \Delta_{b_t}(S),$$

therefore,

$$\begin{aligned} \Delta_{c_t}(S) &\leqslant (-\mathrm{m}(b_t, c_t; t) + \beta_{b_t}(t, \delta) + \beta_{c_t}(t, \delta))^+ + \mathrm{m}(b_t, c_t; t) + \beta_{b_t}(t, \delta) + \beta_{c_t}(t, \delta) \\ &\leqslant 2(\beta_{b_t}(t, \delta) + \beta_{c_t}(t, \delta)) \end{aligned}$$

Put together, we have proved that

$$\Delta_{c_t}(S) \leqslant \beta_{b_t}(t, \delta) + \beta_{c_t}(t, \delta) \text{ and } \Delta_{b_t}(S) \leqslant 2(\beta_{b_t}(t, \delta) + \beta_{c_t}(t, \delta))$$

which again allows to conclude as letting $a_t$ the least explored among $b_t$ and $c_t$,

$$\Delta_{a_t}(S) \leqslant 4\beta_{a_t}(t, \delta)$$

that is $a_t \in W_t(S, I)$ $\qquad \square$

**Lemma C.8.** *If $Z_2(t) > 0$, then, for any $i \in G_t$, there exists $j \in O_t$ such that $\hat{\mu}_{t,i} \prec \hat{\mu}_{t,j}$.*

*Proof.* Recall that arms in $G_t$ cannot yet be confidently classified as feasible or infeasible.

As $Z_2(t) > 0$, and $i \in G_t \cap O_t^c$, from the definition of $Z_2(t)$, it holds that

$$\max_{j \in (F_t \cup G_t) \setminus \{i\}} \mathrm{m}^-(i, j; t) > 0,$$

so there exists $j \in j \in (F_t \cup G_t) \setminus \{i\}$ such that $\mathrm{m}^-(i, j; t) > 0$, in particular, $\hat{\mu}_{t,i} \prec \hat{\mu}_{t,j}$. Introducing

$$\mathfrak{J}_i := \{j \in F_t \setminus \{i\} : \hat{\mu}_{t,i} \prec \hat{\mu}_{t,j}\},$$

we will show that $\mathfrak{J}_i \neq \emptyset$. Let $\Omega_i := \{j \in (F_t \cup G_t) \setminus \{i\} : \hat{\mu}_{t,i} \prec \hat{\mu}_{t,j}\}$ and define $H_i := \mathrm{Par}\,\Omega_i, \hat{\boldsymbol{\mu}}_t$, the Pareto Set of $\Omega_i$ based on the empirical means. By the result above, $\Omega_i$ is non-empty, so $H_i$ is also non-empty.

We claim that $H_i \subset F_t$. Indeed, for $k \in H_t \cap F_t^c$ we have $k \in G_t \cap O_t^c$, then, as $Z_2(t) > 0$, as justified above for $i$, there exists $l \in (F_t \cup G_t) \setminus \{k\}$ such that $\hat{\mu}_{t,k} \prec \hat{\mu}_{t,l}$, however, as $k \in H_t := \mathrm{Par}\,\Omega_i, \hat{\boldsymbol{\mu}}_t$, the above is only possible if $l \notin \Omega_i$. Recalling that $l \in (F_t \cup G_t) \setminus \{k\}$ and $l \neq b_t$ we have $l \in (F_t \cup G_t) \setminus \{b_t\}$ so putting the above displays together, if $H_t \cap F_t^c \neq \emptyset$, there exists $l$ and $k$ such that

$$\hat{\mu}_{t,i} \prec \hat{\mu}_{t,k}; \hat{\mu}_{t,k} \prec \hat{\mu}_{t,l}$$

but $\hat{\mu}_{t,i} \not\prec \hat{\mu}_{t,l}$, which is not possible as Pareto dominance is transitive. Therefore, $H_t \subset F_t$ and since $H_t \neq \emptyset$, $\mathfrak{J}_i \neq \emptyset$. Thus, there exists $j \in F_t \setminus \{b_t\}$ such that $\hat{\mu}_{t,i} \prec \hat{\mu}_{t,j}$. Then, either $j \in O_t$ or there exists $j_2 \in O_t$ such that, $\hat{\mu}_{t,j} \prec \hat{\mu}_{t,j_2}$. In any case, there exists $j_3 \in O_t$ (either $j$ or $j_2$) such that $\hat{\mu}_{t,i} \prec \hat{\mu}_{t,j_3}$. $\square$

### C.2.2. PROOF OF THEOREM 4.4

We recall the theorem below.

**Theorem 4.4.** *Let $f(t, \delta) = \log(\frac{4k_1 K d t^\alpha}{\delta}), g(t, \delta) = 4\log(\frac{4k_1 K 5^d t^\alpha}{\delta})$ with $k_1 > 1 + \frac{1}{\alpha - 1}$ and $\mathcal{D}$ the class of marginally $\sigma$-subGaussian and $\sigma_u$-norm-subGaussian distributions. Then for any $\alpha > 2$ and $\delta \in (0, 1)$, e-cAPE is $\delta$ correct on $\mathcal{D}^K$. Moreover, for all $\nu \in \mathcal{D}^K$ with means $\boldsymbol{\mu} \in \mathcal{I}^K$,*

$$\mathbb{E}_\nu[\tau] \leqslant 128 C_{\mathcal{M}}^*(\nu) \log\left(64 C_{\mathcal{M}}^*(\nu) \left(\frac{4k_1 K d}{\delta}\right)^{1/\alpha}\right)$$

$$+ 4H(\nu, \mathrm{F}^c) + \Lambda_\alpha,$$

*where $H(\nu, \mathrm{F}^c) := \sum_{a \in \mathrm{F}^c \cup O^*} \frac{32\sigma_u^2 \log(5^d/d)}{\eta_a^2}$ and*

$$\Lambda_\alpha \leqslant \frac{2^{\alpha-1}}{4k_1} \sum_{T \geqslant 1} (\log(T) + 1) \left(\frac{f(T, 5^{-d}d) + f(T, 1) + \frac{2}{e}}{T^{\alpha-1}}\right).$$

*Proof.* Let $(S, I) \in \mathcal{M}$ be fixed. Let $T > 0$ be fixed and observe that

$$\min(\tau_\delta, T) \quad \leqslant \quad \left\lceil \frac{T}{2} \right\rceil + \sum_{t=\lceil T/2 \rceil}^{T} \mathbb{1}_{(t > \tau_\delta)}.$$

Then, assuming

$$\mathcal{E}^T := \bigcap_{\lceil T/2 \rceil \leqslant t \leqslant T} \mathcal{E}_t \tag{35}$$

holds, and using Proposition C.3 we have

$$
\begin{aligned}
\min(\tau_\delta, T) &\leqslant \left\lceil \frac{T}{2} \right\rceil + \sum_{t=\lceil T/2 \rceil}^{T} \mathbb{1}_{(t>\tau_\delta)} \\
&\leqslant \left\lceil \frac{T}{2} \right\rceil + \sum_{t=\lceil T/2 \rceil}^{T} \mathbb{1}_{(b_t \in W_t(S,I) \vee c_t \in W_t(S,I))} \\
&\leqslant \left\lceil \frac{T}{2} \right\rceil + \sum_{t=\lceil T/2 \rceil}^{T} \sum_{a=1}^{K} \mathbb{1}_{(b_t=a \vee c_t=a)} \mathbb{1}_{(a \in W_t(S,I))} \\
&= \left\lceil \frac{T}{2} \right\rceil + \sum_{a=1}^{K} \sum_{t=\lceil T/2 \rceil}^{T} \mathbb{1}_{(b_t=a \vee c_t=a)} \mathbb{1}_{(a \in W_t(S,I))} ,
\end{aligned}
$$

then, we decompose the latter sum into three terms by defining the function $R$ for $U \subset [K]$ as

$$
R(U) := \sum_{a \in U}^{K} \sum_{t=\lceil T/2 \rceil}^{T} \mathbb{1}_{(b_t=a \vee c_t=a)} \mathbb{1}_{(a \in W_t(S,I))}
$$

we have for the set $S \subset \mathrm{SubOpt}$, using the definition of $W_t^3(S)$ for $a \in S$:

$$
\begin{aligned}
R(S) &= \sum_{a \in S} \sum_{t=\lceil T/2 \rceil}^{T} \mathbb{1}_{(b_t=a \vee c_t=a)} \mathbb{1}_{(\Delta_a(S) \leqslant 4\beta_a(t,\delta))} \\
&\leqslant \sum_{a \in S} \sum_{t=\lceil T/2 \rceil}^{T} \mathbb{1}_{(b_t=a \vee c_t=a)} \mathbb{1}_{(N_{t,a} \leqslant 32\sigma^2 \Delta_a^{-2}(S) f(t,\delta))} \\
&\leqslant \sum_{a \in S} \sum_{t=\lceil T/2 \rceil}^{T} \mathbb{1}_{(b_t=a \vee c_t=a)} \mathbb{1}_{(N_{t,a} \leqslant 32\sigma^2 \Delta_a^{-2}(S) f(T,\delta))} \quad \text{(as } f \text{ is increasing)}
\end{aligned}
$$

therefore,

$$
R(S) \leqslant \sum_{a \in S} \frac{32\sigma^2}{\Delta_a^2(S)} f(T,\delta). \tag{36}
$$

Proceeding similarly for $I$, we obtain

$$
R(I) \leqslant \sum_{a \in I} \frac{8\sigma_u^2}{\eta_a^2} g(T,\delta), \tag{37}
$$

and for $O^*$, we obtain

$$
R(O^*) \leqslant \sum_{a \in O^*} \max \left( \frac{8\sigma_u^2 g(T,\delta)}{\eta_a^2}, \frac{32\sigma^2 f(T,\delta)}{\Delta_a^2(S)} \right). \tag{38}
$$

Then,

$$
\begin{aligned}
R(O^*) &\leqslant \sum_{a \in O^*} 16 \max \left( \frac{2\sigma_u^2 \log \left( \frac{4k_1 K 5^d T^\alpha}{\delta} \right)}{\eta_a^2}, \frac{2\sigma^2 \log \left( \frac{4k_1 K d T^\alpha}{\delta} \right)}{\Delta_a^2(S)} \right) \\
&\leqslant \sum_{a \in O^*} 16 \max \left( \frac{2\sigma_u^2}{\eta_a^2}, \frac{2\sigma^2}{\Delta_a^2(S)} \right) \log \left( \frac{4k_1 K d T^\alpha}{\delta} \right) + \sum_{a \in O^*} \frac{32\sigma_u^2}{\eta_a^2} \log \left( \frac{5^d/d}{\delta} \right) .
\end{aligned}
$$

Putting together (38), (37) and (36), yields

$$
\begin{aligned}
R([K]) &\leqslant 16 \left( \sum_{a \in O^*} 2 \max \left( \frac{\sigma^2}{\Delta_a(S)^2}, \frac{\sigma_u^2}{\eta_a^2} \right) + \sum_{a \in S} \frac{2\sigma^2}{\Delta_a^2(S)} + \sum_{a \in I} \frac{2\sigma_u^2}{\eta_a^2} \right) f(T, \delta) + H(\nu, I), \\
&= 16 C(\nu, S, I) f(T, \delta) + H(\nu, I),
\end{aligned}
$$

where $H(\nu, I) = \sum_{a \in I \cup O^*} \frac{32\sigma_u^2 \log(5^d/d)}{\eta_a^2}$. Therefore, assuming (35), we have proved that

$$
\min(\tau_\delta, T) < \left\lceil \frac{T}{2} \right\rceil + 16 C(\nu, S, I) f(T, \delta) + H(\nu, I)
$$

Then, taking $T$ such that the RHS is smaller than $T$ would yield that the algorithm stops before $T$. Applying Lemma D.1 with $a = 64\sigma^2 C(\nu, S, I)\alpha$ and $b = (4k_1 K d/\delta)^{1/\alpha}$ yields

$$
T \geqslant 2a \log(ab) \implies a \log(bT) < T,
$$

that is, letting $\widetilde{C}(\nu, S, I) = 64 C(\nu, S, I)\alpha$,

$$
T \geqslant 2\widetilde{C}(\nu, S, I) \log \left( \widetilde{C}(\nu, S, I)(4k_1 K d/\delta)^{1/\alpha} \right) \implies 64 C(\nu, S, I) \log((4k_1 K d/\delta)T) < T \tag{39}
$$

$$
\implies 16 C(\nu, S, I) f(T, \delta) < T/4. \tag{40}
$$

Moreover, we trivially have for $T > 4H(\nu, I)$, $H(\nu, I) < T/4$. Put together with (39), for $T > T^*(S, I) := 2\widetilde{C}(\nu, S, I) \log \left( \widetilde{C}(\nu, S, I)(4k_1 K d/\delta)^{1/\alpha} \right) + 4H(\nu, I)$, we have $\min(\tau_\delta, T) < T$ so the algorithm must have stopped before round $T$. Therefore,

$$
\forall T > T^*(S, I), \tau_\delta > T \implies (\mathcal{E}^T)^c
$$

that is $\mathbb{E}_\nu[\tau_\delta] \leqslant T^*(S, I) + \sum_{t > T^*(S, I)} \mathbb{P}_\nu((\mathcal{E}^t)^c)$. Letting $\Lambda_\alpha := \sum_{T \geqslant 1} \mathbb{P}_\nu((\mathcal{E}^T)^c)$ and as $S, I$ is arbitrary, we have

$$
\mathbb{E}_\nu[\tau_\delta] \leqslant \min_{(S, I) \in \mathcal{M}} T^*(S, I) + \Lambda_\alpha
$$

and from lemma D.3,

$$
\Lambda_\alpha \leqslant \frac{2^{\alpha-1}\delta}{4k_1} \sum_{T \geqslant 1} (\log(T) + 1) \left( \frac{f(T, \delta \cdot d/5^d) + f(T, \delta)}{T^{\alpha-1}} \right).
$$

By definition of the function $f$, we have $f(T, \delta \cdot d/5^d) + f(T, \delta) = f(T, 5^{-d}d) + f(T, 1) + 2\log(1/\delta)$. Using that $\delta \leqslant 1$ and $\delta \log(1/\delta) \leqslant 1/e$ permits to obtain the following upper bound, which is now independent from $\delta$:

$$
\Lambda_\alpha \leqslant \frac{2^{\alpha-1}}{4k_1} \sum_{T \geqslant 1} (\log(T) + 1) \left( \frac{f(T, 5^{-d}d) + f(T, 1) + 2e^{-1}}{T^{\alpha-1}} \right).
$$

Finally, as $x \mapsto x \log(x)$ is increasing on $(1, \infty)$, we have

$$
\mathbb{E}_\nu[\tau_\delta] \leqslant 128 C_{\mathcal{M}}^*(\nu) \log(64 C_{\mathcal{M}}^*(\nu)(4k_1 K d/\delta)^{1/\alpha}) + 4H(\nu, F^c) + \Lambda_\alpha.
$$

$\square$

# D. Concentration Inequalities and Technical Lemmas

We state some concentration inequalities that are used in our proofs.

**Lemma 4.2.** *Under $\mathcal{E}_t$, for all $i, j$ it holds : (i) $|\mathrm{M}(i, j) - \mathrm{M}(i, j; t)| \leqslant \beta_i(t) + \beta_j(t)$ and $|\mathrm{m}(i, j) - \mathrm{m}(i, j; t)| \leqslant \beta_i(t) + \beta_j(t)$, (ii) for any $\mathcal{X} \subset \mathbb{R}^d$, $|\mathrm{dist}(\hat{\mu}_{t,i}, \mathcal{X}) - \mathrm{dist}(\mu_i, \mathcal{X})| \leqslant U_i(t)$.*

*Proof.*

$$
\begin{aligned}
|\mathrm{M}(i,j) - \mathrm{M}(i,j;t)| \;&\leqslant\; \max_{c\leqslant d}\left|\left(\mu_i^c - \mu_j^c\right) - \left(\hat{\mu}_{t,i}^c - \hat{\mu}_{t,j}^c\right)\right| \\
&=\; \max_{c\leqslant d}\left|\left(\mu_i^c - \hat{\mu}_{t,i}^c\right) - \left(\mu_j^c - \hat{\mu}_{t,j}^c\right)\right| \\
&\leqslant\; \beta_i(t) + \beta_j(t),
\end{aligned}
$$

where the last inequality follows under $\mathcal{E}_t$. Further noting that $\mathrm{m}(i,j) = -\mathrm{M}(i,j)$ and $\mathrm{m}(i,j;t) = -\mathrm{M}(i,j;t)$ proves the first point. Letting $\mathcal{X} \subset \mathbb{R}^d$ non-empty we have for all arm $i$

$$
\begin{aligned}
|\mathrm{dist}(\hat{\mu}_{t,i}, \mathcal{X}) - \mathrm{dist}(\mu_i, \mathcal{X})| \;&=\; \left|\sup_{x\in\mathcal{X}}\left[-\|\mu_i - x\|_2\right] - \sup_{x\in\mathcal{X}}\left[-\|\hat{\mu}_{t,i} - x\|_2\right]\right| \\
&\overset{(i)}{\leqslant}\; \sup_{x\in\mathcal{X}}\left|\|\hat{\mu}_{t,i} - x\|_2 - \|\mu_i - x\|\right| \\
&\overset{(ii)}{\leqslant}\; \|\hat{\mu}_{t,i} - \mu_i\| \\
&\leqslant\; U_i(t,\delta)
\end{aligned}
$$

where the last inequality follows from $\mathcal{E}_t$, $(i)$ follows from the inequality

$$
|\sup f - \sup g| \leqslant \sup|f - g|
$$

and $(ii)$ follows from the $|\|x\| - \|y\|| \leqslant \|x - y\|$. $\qquad\square$

**Lemma D.1.** *Let $a, b \geqslant e$. The following statement holds :*

$$
t \geqslant 2a\log(ab) \implies a\log(bt) \leqslant t.
$$

*Proof.* First, note that $t \mapsto \log(bt)/t$ is decreasing on $[e/b, +\infty)$ and observe for $t_0 = 2a\log(ab)$, it holds that

$$
\begin{aligned}
a\log(bt_0) \;&=\; a\log(2ab\log(ab)) \\
&=\; a\log(ab) + a\log(2\log(ab) \\
&<\; 2a\log(ab) \quad (\text{as } \log(x) < x/2) \\
&=\; t_0
\end{aligned}
$$

therefore, $a\frac{\log(t_0)}{t_0} < 1$ and as the function is decreasing

$$
t \geqslant t_0 \implies a\frac{\log(bt)}{t} \leqslant a\frac{\log(bt_0)}{t_0} < 1.
$$

$\qquad\square$

We recall the following result of Katz-Samuels & Scott (2019), which has been used in the proof of the lower bound.

**Lemma D.2** (Lemma F.2 of Katz-Samuels & Scott (2019))**.** *Let $x \in \mathbb{R}^d$ and $S \subset \mathbb{R}^d$ be a closed non-empty set. Then there exists $y \in S$ such that $\|x - y\| = \mathrm{dist}(x, S)$.*

**Lemma D.3.** *Let $f(t,\delta) = \log(\frac{4k_1 K dt^\alpha}{\delta})$, $g(t,\delta) = 4\log(\frac{4k_1 K 5^d t^\alpha}{\delta})$ with $k_1 > 1 + \frac{1}{\alpha-1}$. Then for all $T$ we have*

$$
\mathbb{P}_\nu((\mathcal{E}^T)^c) \leqslant \frac{\delta}{4k_1}\left(\frac{T}{2}\right)^{1-\alpha}(\log(T)+1)f(T,\delta) + \frac{\delta}{4k_1}\left(\frac{T}{2}\right)^{1-\alpha}(\log(T)+1)f(T,\delta\cdot d/5^d),
$$

*where $\mathcal{E}^T := \bigcap_{\lceil T/2\rceil \leqslant t \leqslant T}\mathcal{E}_t$. And, it holds that $\mathbb{P}_\nu((\mathcal{E})^c) \leqslant \delta$.*

*Proof.* Let $\overline{X}_s$ denote the sample of a centered $\sigma$-subgaussian variable $X$ based on $s$ *i.i.d* observations. We have by Theorem 1 of Garivier (2013),

$$
\begin{aligned}
\mathbb{P}_\nu \left( \exists s \leqslant t : \sqrt{s}|\overline{X}_s| > \sqrt{2\sigma^2 f(t,\delta)} \right) &\leqslant 2e\lceil f(t,\delta) \log(t) \rceil \exp(-f(t,\delta)) \\
&\leqslant 2ef(t,\delta)(\log(t) + 1) \exp(-f(t,\delta))
\end{aligned}
$$

and by the proof of Proposition 5.9 in Kaufmann (2014), it holds that for any $T$

$$
\sum_{t=1}^{T} 2ef(t,\delta)(\log(t) + 1) \exp(-f(t,\delta)) \leqslant Tf(T,\delta)(\log(T) + 1) \exp(-f(T/2,\delta))
$$

therefore,

$$
\begin{aligned}
\mathbb{P}_\nu \left( \exists t \leqslant T, \exists s \leqslant t : \sqrt{s}|\overline{X}_s| > \sqrt{2\sigma^2 f(t,\delta)} \right) &\leqslant Tf(T,\delta)(\log(T) + 1) \exp(-f(T/2,\delta)) \\
&= \frac{\delta}{4k_1 K d} \left( \frac{T}{2} \right)^{1-\alpha} (\log(T) + 1) f(T,\delta).
\end{aligned}
$$

Thus, by union bounds over $K, d$ we derive for vector valued subgaussian random variables,

$$
\mathbb{P}_\nu \left( \exists t \leqslant T, \exists i \in [K] : \|\hat{\mu}_{t,i} - \mu_i\|_\infty > \sqrt{2\sigma^2 f(t,\delta)/N_{t,i}} \right) \leqslant \frac{\delta}{4k_1} \left( \frac{T}{2} \right)^{1-\alpha} (\log(T) + 1) f(T,\delta).
$$

For the second part of the event $\mathcal{E}^T$, note that by Lemma F.4 of Katz-Samuels & Scott (2019), for any $\mathbb{R}^d$ centered valued random vector $X$

$$
\mathbb{P}_\nu(\|X\|_2 > g(t,\delta)) \leqslant \mathbb{P}_\nu \left( \exists z \in \mathcal{N}(\varepsilon) : \frac{1}{1-\varepsilon} X^\mathsf{T} z > g(t,\delta) \right)
$$

where $\mathcal{N}(\varepsilon)$ is a subset of the unit $\mathbb{R}^d$ sphere and is an $\varepsilon$-net or $\varepsilon$-cover of the unit $\mathbb{R}^d$ sphere with $|\mathcal{N}(\varepsilon)| \leqslant (2/\varepsilon + 1)^d$ (cf Lemma F.5 of Katz-Samuels & Scott (2019)). Then, noting that $X^\mathsf{T} z$ is $\sigma_u$-norm-subgaussian and taking $\varepsilon = 1/2$, we have

$$
\mathbb{P}_\nu(\|X\|_2 > U(t,\delta)) \leqslant \mathbb{P}_\nu \left( \exists z \in \mathcal{N}(1/2) : |X^\mathsf{T} z| > \sqrt{2\sigma_u^2 f(t, \delta \cdot d/5^d)} \right).
$$

Thus, by what precedes and applying a union bound over $\mathcal{N}(1/2)$ and $[K]$ we have

$$
\mathbb{P}_\nu \left( \exists t \leqslant T, \exists i \in [K] : \|\hat{\mu}_{t,i} - \mu_i\|_2 > \underbrace{\sqrt{2\sigma_u^2 f(t, \delta \cdot d/5^d)/N_{t,i}}}_{U_i(t,\delta)} \right) \leqslant \frac{\delta}{4k_1} \left( \frac{T}{2} \right)^{1-\alpha} (\log(T) + 1) f(T, \delta \cdot d/5^d),
$$

where we used the fact that $|\mathcal{N}(1/2)| \leqslant 5^d$. Therefore,

$$
\mathbb{P}_\nu((\mathcal{E}^T)^c) \leqslant \frac{\delta}{4k_1} \left( \frac{T}{2} \right)^{1-\alpha} (\log(T) + 1) f(T,\delta) + \frac{\delta}{4k_1} \left( \frac{T}{2} \right)^{1-\alpha} (\log(T) + 1) f(T, \delta \cdot d/5^d).
$$

It remains to prove the last claim of the Lemma. By Hoeffding's sub-gaussian concentration (cf e.g. Lattimore & Szepesvári (2020)), we have

$$
\begin{aligned}
\mathbb{P}_\nu \left( \exists t \geqslant 1, \exists i \in [K], \exists c \in [d] : |\hat{\mu}_{t,i}^c - \mu_i^c| > \beta_i(t,\delta) \right) &\leqslant 2Kd \sum_{t \geqslant 1} \exp(-f(t,\delta)) \\
&\leqslant \frac{\delta}{2k_1} \sum_{t \geqslant 1} \frac{1}{t^\alpha},
\end{aligned}
$$

then, remark that $\sum_{t\geqslant 1}\frac{1}{t^\alpha}\leqslant\int_1^\infty t^{-\alpha}dt=\frac{1}{\alpha-1}$ and $k_1>1+\frac{1}{\alpha-1}$, so that

$$\mathbb{P}_\nu\left(\exists t\geqslant 1,\exists i\in[K],\exists c\in[d]:|\hat{\mu}_{t,i}^c-\mu_i^c|>\beta_i(t,\delta)\right)\leqslant\frac{\delta}{2}.$$

Showing that

$$\mathbb{P}_\nu\left(\exists t\geqslant 1,\exists i\in[K],\|\hat{\mu}_{t,i}-\mu_i\|_2>U_i(t,\delta)\right)\leqslant\frac{\delta}{2},$$

follows similarly by using the covering technique explained above. Combining the two displays we conclude that $\mathbb{P}_\nu((\mathcal{E})^c)\leqslant\delta$. $\qquad\square$

# E. Gap-Based Lower Bound on the Sample Complexity of e-cPSI

In this section, we prove a lower bound on the sample complexity of any $\delta$-correct algorithm for the constrained PSI problem. As this is a pure exploration problem with multiple correct answers, it might be tempting to apply Theorem 1 of Degenne & Koolen (2019). However, their result does not directly apply to our setting as the authors studied the regime $\delta\to 0$ and considered answers that are singletons of $[K]$ while we consider partitions of $[K]$.

We state below the result we will prove in this section. We denote by $\widetilde{\mathcal{D}}$ a family of bandit instances that will become explicit by the end of the section.

**Theorem E.1.** *Let $P\coloneqq\{x\in\mathbb{R}^d\mid Ax\prec b\}$ and let $\nu\in\widetilde{\mathcal{D}}^K$ with means $\boldsymbol{\mu}\in\mathcal{I}^K$. The following holds for any $\delta$-correct algorithm*

$$\mathbb{E}_\nu[\tau]\min_{(S',I')\in\mathcal{M}}C(\nu,S',I')\frac{g(\delta)}{4},\text{where }g(\delta)\coloneqq\mathrm{kl}((1-\delta)/2,\delta)\geqslant\frac{(1-\delta)}{2}\log(1/\delta)-\log(2).$$

*In particular,*

$$\liminf_{\delta\to 0}\frac{\mathbb{E}_\nu[\tau]}{\log(1/\delta)}\geqslant\frac{C_\mathcal{M}^*(\nu)}{8}.$$

In what follows, we consider $\mathcal{A}_\delta$, a $\delta$-correct algorithm (cf Definition 3.3) and $\boldsymbol{\mu}$, a bandit instance, fixed and unknown to the algorithm. We assume that $\tau<\infty$ almost surely (otherwise the lower bound trivially holds). Let $\widehat{O},\widehat{S},\widehat{I}$ be the recommendation of the algorithm after $\tau$ (random) steps and $\mathcal{F}_\tau$ the natural filtration of the stochastic process $(A_t,X_t)$ stopped at round $\tau$. Since $\mathcal{A}$ is $\delta$-correct, the following statement holds

$$\mathbb{P}_\nu\left(\widehat{O}=O^*\text{ and }(\widehat{S},\widehat{I})\in\mathcal{M}\right)\geqslant 1-\delta. \tag{41}$$

We define

$$\widetilde{S}\coloneqq\{i\in[K]\mid\mu_i\in P\text{ and }\exists j\in O^*\mid\mu_i\prec\mu_j\}$$

the set of arms that are feasible but sub-optimal. Note that alternatively, $\widetilde{S}=\mathrm{SubOpt}\cap\mathrm{F}$. Similarly, we introduce

$$\widetilde{I}\coloneqq\{i\in[K]\mid\mu_i\notin P\text{ and }\nexists j\in O^*\mid\mu_i\prec\mu_j\},$$

the set of arms that are unsafe but not uniformly worse than any other optimal arm, which rewrites as $\widetilde{I}=\mathrm{F}^c\cap\mathrm{SubOpt}^c$. In particular, $\widetilde{S}\cap\widetilde{I}=\emptyset$ and it can be observed that for any valid answer $(S,I)\subset\mathcal{M}$, $\widetilde{S}\subset S$ and $\widetilde{I}\subset I$. Thus, for any $\delta$-correct algorithm it holds that

$$\forall i\in\widetilde{I},\ \mathbb{P}_\nu(i\in\widehat{I})\geqslant 1-\delta\quad\text{and}\quad\forall i\in\widetilde{S},\ \mathbb{P}_\nu(i\in\widehat{S})\geqslant 1-\delta. \tag{42}$$

Then by definition, every arm of $\mathcal{B}\coloneqq(O^*)^c\cap(\widetilde{S}\cup\widetilde{I})^c$ can be (correctly) classified either in $\widehat{S}$ or $\widehat{I}$, resulting in $2^{|\mathcal{B}|}$ correct answers, which can be up to $2^{K-1}$ correct answers.

### E.1. A high probability correct answer for $\mathcal{A}_\delta$

We describe a particular correct answer under bandit $\nu$ with mean $\boldsymbol{\mu}$ that is very likely to be recommended by $\mathcal{A}_\delta$.

We build on $(\widetilde{S}, \widetilde{I})$ to construct a partition with $(S, I)$ for which a property similar to Equation (42) holds for every arm of $(S, I)$. To construct such instance, note that as $\mathcal{A}_\delta$ is $\delta$-correct, for any $i \in \mathcal{B} = (O^*)^c \cap (\widetilde{S} \cup \widetilde{I})^c$, it holds that $\mathbb{P}_\nu(i \in \widehat{S} \cup \widehat{I}) \geqslant 1 - \delta$, which yields

$$\max\left(\mathbb{P}_\nu(i \in \widehat{S}), \mathbb{P}_\nu(i \in \widehat{I})\right) \geqslant \frac{1-\delta}{2}. \tag{43}$$

Then, we define $S, I$ as

$$S := \widetilde{S} \cup \left\{i \in C : \mathbb{P}_\nu(i \in \widehat{S}) > \mathbb{P}_\nu(i \in \widehat{I})\right\}$$

and we similarly define the set

$$I := \widetilde{I} \cup \left\{i \in C \mid \mathbb{P}_\nu(i \in \widehat{S}) \leqslant \mathbb{P}_\nu(i \in \widehat{I})\right\}.$$

By construction, $(S, I)$ satisfies $S \cap I = \emptyset$ and $S \cup I = (O^*)^c$, i.e $(S, I)$ is a valid answer. Moreover, one can verify that

a) $\mathbb{P}_\nu(\widehat{O} = O^*) \geqslant 1 - \delta$

b) for all $i \in S$, $\mathbb{P}_\nu(i \in \widehat{S}) \geqslant \frac{1-\delta}{2}$ and

c) for all $i \in I$, $\mathbb{P}_\nu(i \in \widehat{I}) \geqslant \frac{1-\delta}{2}$;

that is this particular $(O^*, S, I)$ is a likely response for the algorithm $\mathcal{A}_\delta$. Using a change of distribution lemma will allow to derive a lower bound on the number of pulls of each arm for $\mathcal{A}_\delta$ to identify $(S, I)$ as a correct answer. In the sequel, we restrict ourselves to bandit with multi-variate normal arms and identity covariance. Therefore, each arm will be identified with its mean vector.

**Lemma E.2** ((Kaufmann et al., 2016)). *For all bandit models $\nu, \nu'$ and for any $\mathcal{F}_\tau$-measurable event $\mathcal{E}$,*

$$\sum_{i=1}^{K} \mathbb{E}_\nu[N_{\tau,i}] \, \mathrm{KL}(\nu_i, \nu'_i) \geqslant \mathrm{kl}(\mathbb{P}_\nu(\mathcal{E}), \mathbb{P}_{\nu'}(\mathcal{E})),$$

*where $\mathrm{KL}(\nu_i, \nu'_i)$ is the Kullback-Leibler divergence between distributions $\nu_i, \nu'_i$ and $\mathrm{kl}$ is the relative binary entropy.*

### E.2. Change of distribution

The idea now is to apply the classical change of distribution lemma to different well chosen instances of bandit. We define instances with $\nu^1, \ldots, \nu^K$ with (matrix of) means (resp.) $\boldsymbol{\lambda}^1, \ldots, \boldsymbol{\lambda}^K$ such that $\boldsymbol{\lambda}^i := (\lambda_k^i)_{1 \leqslant k \leqslant K}$ and $\boldsymbol{\lambda}^i$ differs from $\boldsymbol{\mu}$ only in the mean of arm $i$ i.e., the bandits $\nu$ and $\nu^i$ are identical except $\mathbb{E}_{X \sim \nu_i^i}[X] = \lambda_i^i \neq \mu_i = \mathbb{E}_{X \sim \nu_i}[X]$. For such instances, applying Lemma E.2 yields

$$\mathbb{E}_\nu[N_{\tau,i}] \, \mathrm{KL}(\nu_i, \nu_i^i) \geqslant \sup_{\mathcal{E} \in \mathcal{F}_\tau} \mathrm{kl}(\mathbb{P}_\nu(\mathcal{E}), \mathbb{P}_{\nu^i}(\mathcal{E})) \tag{44}$$

where in our case, $\mathrm{KL}(\mu_i, \lambda_i^i)$ is the Kullback-Leibler divergence between multivariate normals of means $\nu_i, \nu_i^i$ and identity covariance. $\mathrm{kl}(x, y) := x \log(x/y) + (1-x) \log((1-x)/(1-y))$, and noting that $x \mapsto \mathrm{kl}(x, y)$ is increasing on $\{x > y\}$ and decreasing on $\{x < y\}$, making an event likely under $\nu$ unlikely under $\nu^i$ will increase the RHS of Equation (44). Going back to the instance $(S, I)$ constructed earlier, we introduce the event

$$\mathcal{E}_i := \begin{cases} \{i \in \widehat{S}\} & \text{if } i \in S \\ \{i \in \widehat{I}\} & \text{if } i \in I, \\ \{\widehat{O} = O^*\} & \text{if } i \in O^*. \end{cases}$$

We recall that $(O^*, S, I)$ forms a partition of $[K]$. From the definition of $(S, I)$ we have shown that for any $\delta$-correct algorithm,

$$\mathbb{P}_\nu(\mathcal{E}_i) \geqslant \begin{cases} 1 - \delta & \text{if } i \in O \\ \frac{1-\delta}{2} & \text{else,} \end{cases}$$

that is such an event is very likely under bandit $\mu$. We will choose each instance $\lambda^i$ such that $\mathcal{E}_i$ is unlikely under bandit $\lambda^i$, i.e we assume that $\lambda^i$ is chosen such as for all $i$

$$\mathbb{P}_{\nu^i}(\mathcal{E}_i) \leqslant \delta. \tag{45}$$

We now discuss the choice of $\lambda^i$ depending on $i$ to ensure such property. Thus, assuming this property holds, we have for all arm $i$

$$\mathbb{E}_\nu[N_{\tau,i}] \, \mathrm{KL}(\nu_i, \nu_i^i) \geqslant \mathrm{kl}\left(\frac{1-\delta}{2}, \delta\right),$$

which using the KL formula for multivariate normal yields

$$\mathbb{E}_\nu[N_{\tau,i}] \geqslant \frac{2}{\|\mu_i - \lambda_i^i\|^2} \underbrace{\mathrm{kl}\left(\frac{1-\delta}{2}, \delta\right)}_{g(\delta)}.$$

**Case 1:** $i \in I$ **an unsafe arm.** In this case, $\mu_i \notin P$. Since $P$ is a closed convex set, by Hilbert projection onto closed convex sets, there exists (cf Lemma D.2) $z_i \in P$ such that $\|\mu_i - z_i\|_2 = \mathrm{dist}(\mu_i, P)$. We now define $\lambda_i^i = z_i$. In this bandit $\lambda_i$, $i$ is now a safe arm so, as $\mathcal{A}_\delta$ is a $\delta$-correct algorithm it holds that

$$\mathbb{P}_{\nu^i}(i \in \widehat{I}) \leqslant \delta \quad \text{i.e} \quad \mathbb{P}_{\nu^i}(\mathcal{E}_i) \leqslant \delta,$$

which by further noting that $\|\mu_i - \lambda_i^i\|_2 = \mathrm{dist}(\mu_i, P) = \eta_i$ results in

$$\mathbb{E}_\nu[N_{\tau,i}] \geqslant \frac{2}{\eta_i^2} g(\delta).$$

**Case 2:** $i \in S$ **a sub-optimal arm.** Note that in this case, there exists a subset $\Omega_i \subset O^*$ such that $\mu_i \prec \mu_j$ for all $j \in \Omega_i$: it is the set of optimal arms that dominate $\mu_i$. For any $j \in \Omega_i$ we define

$$c_j = \underset{c}{\mathrm{argmin}} \, [\mu_j(c) - \mu_i(c)]$$

the coordinate for which the margin between $i$ and $j$ is the lowest. In particular, $\mathrm{m}(i, j) = \mu_j(c_j) - \mu_i(c_j)$. Moreover, for any $\varepsilon > 0$, the vector defined by

$$\tilde{\mu}_i(c) = \begin{cases} \mu_i(c) + (1 + \varepsilon) \, \mathrm{m}(i, j) & \text{if } c = c_j \\ \mu_i(c) & \text{else} \end{cases}$$

is not dominated by $\mu_j$. Now let us define the vector $s_i$ such that for any $c \in [d]$,

$$s_i(c) := \max_{j \in \Omega_i : c_j = c} \mathrm{m}(i, j) \tag{46}$$

with the convention that $\max_\emptyset = 0$. We argue that defining

$$\lambda_i^i := \mu_i + (1 + \varepsilon) s_i,$$

we have

$$\mathbb{P}_{\nu^i}(i \in \widehat{S}) \leqslant \delta.$$

To see this, note that as only the mean of arm $i$ has changed between bandits $\nu$ and $\nu^i$, the set of feasible and infeasible arms of $[K] \backslash \{i\}$ under $\nu$ and $\nu^i$ are the same. Moreover, the change of distribution ensures that under the bandit $\nu^i$, $i$ is not dominated by any feasible arm. Therefore, $\mathbb{P}_{\nu^i}(i \in \hat{S}) \leqslant \delta$. So applying what precedes and letting $\varepsilon \to 0$ yield

$$\mathbb{E}_\nu[N_{\tau,i}] \geqslant \frac{2}{\|s_i\|^2} g(\delta). \tag{47}$$

**Case 3:** $i \in O^*$ **the optimal set.** We recall that arms in $O^*$ are both feasible and Pareto optimal among feasible arms. We then build alternative instances where arm $i \in O^*$ is either made infeasible or sub-optimal among feasible arms under bandit $\nu^i$.

Letting $\eta_i = \text{dist}(\mu_i, P^c) = \text{dist}(\mu_i, \partial P)$ (as $\mu_i \in P$, cf (16)) and since $\partial P$ is a closed, using Lemma F.2 of Katz-Samuels & Scott (2019) (recalled in Lemma D.2), there exists $z_i \in \partial P$ such that $\text{dist}(\mu_i, \partial P) = \|\mu_i - z_i\|_2 = \eta_i$. Then, note that, as $z_i \notin P^\circ$ (the interior of $P$), for all $\varepsilon > 0, \exists z_i^\varepsilon \in P^c$ such that $\|z_i^\varepsilon - z_i\| \leqslant \varepsilon$. Then, letting $\lambda_i^i = z_i^\varepsilon$ (for some $\varepsilon$ fixed). It comes that $i$ is infeasible under $\lambda^i$, so that $\mathbb{P}_{\nu^i}(\mathcal{E}_i) \leqslant \delta$. Therefore, we have

$$\mathbb{E}_\nu[N_{\tau,i}] \geqslant \frac{2}{\|\mu_i - z_i^\varepsilon\|^2} g(\delta)$$
$$= \frac{2}{\|\mu_i - z_i + (z_i - z_i^\varepsilon)\|^2} g(\delta).$$

Letting $\varepsilon \to 0$ yields

$$\mathbb{E}_\nu[N_{\tau,i}] \geqslant \frac{2}{\|\mu_i - z_i\|^2} g(\delta) = \frac{2}{\eta_i^2} g(\delta).$$

Alternative change of distribution will prove the dependency on the other quantities involved in the gaps. Assume the gap of $i$ is attained for an arm $j \in O^*$ with $\Delta_i = \text{M}(i,j)$. In this case, $j$ is close to dominating $i$ so that decreasing $i$ will make it dominated by $j$, as result, $i$ becomes either feasible but dominated or non-feasible, i.e $i \notin O^*$ in both cases. We define in this case

$$\lambda_i^i := \mu_i - (1+\varepsilon)(\mu_i - \mu_j)_+$$

where $(x)_+$ is defined component-wise for $x \in \mathbb{R}^d$. Then it can be easily checked that under bandit $\nu^i$, arm $i$ is now dominated by $j$ and $j$ is still feasible. Therefore, $\mathbb{P}_{\nu^i}(\mathcal{E}_i) \leqslant \mathbb{P}_{\nu^i}(i \in \widehat{O}) \leqslant \delta$. So that proceeding as before we prove that

$$\mathbb{E}_\nu[N_{\tau,i}] \geqslant \frac{2}{\|(\mu_i - \mu_j)_+\|^2} g(\delta). \tag{48}$$

Similarly, if we had $\Delta_i = \text{M}(j,i)$ for some $j \in O^*$. Then, increasing $i$ will make it dominate $j$. Thus, defining

$$\lambda_i^i := \mu_i + (1+\varepsilon)(\mu_j - \mu_i)_+,$$

on can observe that arm $j$ is now dominated by $i$ under bandit $\nu^i$. Note that $j$ is still feasible in bandit $\nu^i$ (since its mean has not changed). So two things may occur. Either $i$ is still feasible under $\nu^i$, in which case $j$ is feasible and sub-optimal in bandit $\nu^i$ or $i$ is now infeasible under bandit $\nu^i$ so that $i$ is no longer an optimal arm. In both cases, the optimal set has changed, so that $\mathbb{P}_{\nu^i}(\mathcal{E}_i) \leqslant \delta$. Reasoning as in the previous cases,

$$\mathbb{E}_\nu[N_{\tau,i}] \geqslant \frac{2}{\|(\mu_j - \mu_i)_+\|^2} g(\delta). \tag{49}$$

Therefore, by defining the quantity we have proved that

$$\mathbb{E}_\nu[N_{\tau,i}] \geqslant \frac{2}{\min(\widetilde{\delta}_i^+, \eta_i)^2} g(\delta),$$

where

$$\widetilde{\delta}_i^+ = \min_{j \in O^* \setminus \{i\}} \min(\|(\mu_j - \mu_i)_+\|_2, \|(\mu_i - \mu_j)_+\|_2). \tag{50}$$

Note that this is a strictly positive quantity as $i \in O^*$ and from the definition of non-dominance for all $j \in O^*$.

### E.3. Relation to the gaps in PSI

We recall the expressions of the PSI gaps (w.r.t $O^*$) as introduced in Auer et al. (2016). For a sub-optimal arm $i \in S$,

$$\Delta_i := \Delta_i^* := \max_{j \in O^*} \text{m}(i,j), \tag{51}$$

For an optimal arm $i \in O^*$,
$$\Delta_i(S \cup O^*) := \min(\delta_i^+, \delta_i^-(S))$$
where
$$\delta_i^+ := \min_{j \in O^* \setminus \{i\}} \min(\mathrm{M}(i,j), \mathrm{M}(j,i)) \quad \text{and} \quad \delta_i^-(S) := \min_{j \in S}[(\mathrm{M}(j,i))^+ + \Delta_j],$$
and $\mathrm{M}(i,j) = \max_{c \in [d]}[\mu_i(c) - \mu_j(c)]$. Since $O^*$ is fixed, we ease notation and write $\Delta_i(S \cup O^*) = \Delta_i(S)$ for all $i \in O^*$.

As justified in Remark 18 of Auer et al. (2016), $\delta_i^-(S)$, can be compensated by both $\delta_i^+(S)$ and $\Delta_j^*$ for some arms $j \in \mathrm{SubOpt}$ (in our case). So we focus on matching the gaps in $\delta_i^+$ for $i \in O^*$. Moreover, from the definition of $\mathrm{M}(i,j)$ for $i, j \in O^*$ (and from the definition of Pareto dominance), it can be easily checked that for any $i \in O^*$,
$$\delta_i^+ := \min_{j \in O^* \setminus \{i\}} \min(\|(\mu_j - \mu_i)_+\|_\infty, \|(\mu_i - \mu_j)_+\|_\infty). \tag{52}$$

Intuitively, (in the case of independent objectives), (50) suggests to measure the distance between Pareto optimal arms in Euclidean whereas (52) measures in sup norm. As these quantities are computed over Pareto optimal arms, it can be checked that for $d = 2$, both terms are equal. In the worst case, the discrepancy between $\delta_i^+$ and $\widetilde{\delta}_i^+$ is at most $\sqrt{d}$. However, as we discuss later, there are classes of PSI/constrained PSI instances where these quantities are equal.

Similarly, one can check that for a sub-optimal arm, the discrepancy between $\Delta_i^*$ and $\|s_i\|^2$ (46) is at most $\min(\sqrt{|\Omega_i|}, \sqrt{d})$, where we recall that
$$\Omega_i := \{j \in O^* : \mu_i \prec \mu_j\}.$$

Thus, in instances where $\Omega_i$ is singleton, that is each of $\mathrm{SubOpt}$ is dominated by a unique feasible optimal arm, $\|s_i\|_2$ and $\Delta_i^*$ are equal. To summarize, we have been justifying that under the following,
$$\forall i \in O^*, \widetilde{\delta}_i^+ = \delta_i^+ \quad \text{and} \quad \forall i \in \mathrm{SubOpt}, \|s_i\|_2 = \Delta_i^* \tag{53}$$

combined with the results of sub-section E.2 we would have
$$\mathbb{E}_\nu[N_{\tau,i}]/(2g(\delta)) \gtrsim \begin{cases} \frac{1}{\eta_i^2} & \text{if } i \in I \\ \frac{1}{(\Delta_i^*)^2} & \text{if } i \in S \\ \frac{1}{\min(\eta_i, \Delta_i(S))^2} & \text{if } i \in O^*. \end{cases} \tag{54}$$

Then, as $\mathbb{E}_\nu[\tau] = \sum_{i=1}^K \mathbb{E}_\nu[N_{\tau,i}]$,
$$\mathbb{E}_\nu[\tau] \geqslant C(\nu, S, I)g(\delta).$$
where we recall that
$$C(\nu, S, I) = \sum_{i \in O^*} \frac{2}{\min(\Delta_i(O^* \cup S), \eta_i)^2} + \sum_{i \in S} \frac{2}{(\Delta_i(O^* \cup S))^2} + \sum_{i \in I} \frac{2}{\eta_i^2}$$

Finally observing that $(S, I)$ is an arbitrary correct response yields
$$\mathbb{E}_\nu[\tau] \gtrsim \min_{(S', I') \in \mathcal{M}} C(\nu, S', I')g(\delta), \tag{55}$$

and observe that $g(\delta) = \mathrm{kl}((1-\delta)/2, \delta) \geqslant \frac{(1-\delta)}{2} \log(1/\delta) - \log(2)$ (as $\mathrm{kl}(x,y) \geqslant x \log(1/y) - \log(2)$).

### E.4. Conclusion

We now give examples of families of instances constructed to satisfy (53). This includes, e.g, instances where the means vectors are close in many objectives except a few (2 or 3). To give more intuition, we build some examples below. We let $\widetilde{\mathcal{D}}$ be the family of multivariate normals with identity covariance and whose mean vector $\mu_1, \ldots, \mu_K$ is as below.

Let $\alpha > 0$ and let $\mu_0 \in P$, and $d \geqslant 1$ (arbitrary). Define $\mu_1 = \mu_0$ and for any $1 < i \leqslant K$, $\mu_k^1 = \mu_{k-1}^1 + \alpha$ and $\mu_k^2 = \mu_{k-1}^2 - \alpha$ and $\mu_k^c = \mu_0^c$ for all $c \notin \{1, 2\}$. Such sequences of vectors are non-dominated by each other and direct computation yields for any $i \in [K]$
$$\widetilde{\delta}_i^+ = 2\alpha \quad \text{and} \quad \delta_i^+ = \alpha.$$

Thus, for $i \in O^*$ the discrepancy between $\widetilde{\delta}_i^+$ and $\delta_i^+$ is 2, irrespective of the dimension. As for such instances, for any $P$, we either have $i \in O^*$ or $\mu_i \notin P$, so, (55) is satisfied with an extra multiplicative constant $1/4$.

Put together, on this set of instances we have

$$\mathbb{E}_\nu[\tau] \gtrsim \min_{(S',I')\in\mathcal{M}} C(\nu, S', I')\frac{g(\delta)}{4}, \tag{56}$$

which gives the claimed result as

$$\liminf_{\delta\to 0} \frac{\mathbb{E}_\nu[\tau]}{\log(1/\delta)} \geqslant \frac{C^*_\mathcal{M}(\nu)}{8}.$$

# F. Racing Algorithm for Constrained Pareto Set Identification

We describe the pseudocode of the racing algorithm, and we comment on its performance. As for racing algorithms, an active set is initialized with all the arms, and the algorithm uses confidence bounds to sequentially discard some arms. The reasons to discard an arm are :

1) it is confidently infeasible

2) it confidently dominated by another arm that itself is confidently feasible

3) it is confidently optimal (in the Pareto Set and feasible) and with high confidence does not dominate another active arm.

Three sets $\widehat{O}, \widehat{S}, \widehat{I}$ are initialized empty, and when discarded, depending on the reason for discarding, it is added to one of them.

---

**Algorithm 3** Racing for constrained Pareto Set Learning

---

$\quad$ *Initialize*: $A_1 \leftarrow \emptyset, \widehat{O} \leftarrow \emptyset, \widehat{S} \leftarrow \emptyset, \widehat{I} \leftarrow \emptyset$
$\quad$ **for** $t = 1, 2, \dots$ **do**
$\quad\quad$ **if** $A_t = \emptyset$ **then**
$\quad\quad\quad$ **break** and **return** $(\widehat{O}, \widehat{S}, \widehat{I})$
$\quad\quad$ **end if**
$\quad\quad$ Pull each active arm once
$\quad\quad$ Compute $\gamma_k(t)$ for each active
$\quad\quad \widehat{I} \leftarrow \widehat{I} \cup \{i \in A_t \mid \hat{\mu}_{t,i} \notin P \text{ and } \gamma_i(t) > 0\}$
$\quad\quad \widehat{S} \leftarrow \widehat{S} \cup \{i \in A_t \mid \exists j \in A_t \text{ s.t. } \hat{\mu}_{t,j} \in P, \gamma_j(t) > 0 \text{ and } \mathrm{m}^-(i,j;t) > 0\}$
$\quad\quad p_t^1 \leftarrow \{i \in A_t \mid \forall j \in A_t\backslash(\widehat{S}\cup\widehat{I}), \mathrm{M}^-(i,j;t) > 0\}$
$\quad\quad p_t^2 \leftarrow \{i \in p_t^1 \mid \forall j \in A_t\backslash(\widehat{S}\cup\widehat{I}), \mathrm{M}^-(j,i;t) > 0\}$
$\quad\quad \widehat{O} \leftarrow \widehat{O} \cup \{i \in p_t^2 \mid i\hat{\mu}_{t,i} \in P \text{ and } \gamma_i(t) > 0\}$
$\quad\quad A_{t+1} \leftarrow A_t\backslash(\widehat{O}\cup\widehat{S}\cup\widehat{I})$
$\quad\quad t \leftarrow t + 1$
$\quad$ **end for**

---

The proof of the correctness of this algorithm follows the same lines as for e-cAPE using concentration inequalities on the quantities $\mathrm{m}, \mathrm{M}$.

**Limitations of Racing for cPSI** One major limitation of this racing algorithm for constrained PSI is linked to the second reason for discarding an arm (as listed above). Indeed, assume $k \in \mathrm{F}\backslash O^*$. As the algorithm is $\delta$-correct, on most runs, $k$ will be added to $\widehat{S}$. So before discarding, we ensure that it is dominated by an arm that itself is already known feasible. This is sub-optimal for an algorithm for constrained PSI. To see this, let us build an instance such that $\mathrm{F} = [K]$ and $O^* = \{1\}$ (w.l.o.g). Further assume that $\mathrm{Par}([K]\backslash\{1\}, \boldsymbol{\mu}) = [K]\backslash\{1\}$, i.e., the remaining arms are not dominated by each other. In this instance, all the arms are feasible, and only 1 is Pareto optimal. Therefore, before correctly removing any arm $k \neq 1$,

the algorithm will ensure that it is dominated by 1 and 1 is confidently feasible. Thus, all the arms would be pulled at least until 1 appears confidently feasible. The sample complexity of the algorithm on this instance will thus scale with

$$(2K\sigma_u^2/\eta_1^2) + \sum_{i \neq 1} \frac{2\sigma^2}{\Delta_i^2([K])}$$

while the sample complexity of cAPE will scale with

$$\frac{2\sigma^2}{\Delta_i^2([K])} + \frac{2\sigma_u^2}{\eta_1^2}.$$

This observation is further confirmed empirically in Appendix G, where both algorithms are benchmarked on similar instances.

# G. Additional Experiments

In this section, we present the results of some additional experiments and further discuss the computational complexity of our algorithms. All the algorithms were mainly implemented in `python 3.10` with some functions in `cython` for faster execution. The experiments were run on an ARM64 8GB RAM/8 core/256GB disk storage computer.

## G.1. Experiments on complex polyhedra

We evaluate our algorithm on different polyhedra in $\mathbb{R}^3$: an ordering polyhedron $P_1 := \{x \in \mathbb{R}^3 : x_i \leqslant x_{i+1}, i \in [2]\}$, a simplex $P_2 := \{x \in \mathbb{R}^3 : x_i \geqslant 0 \text{ and } \sum_i x_i \leqslant 1\}$ and finally a cube $P_3 := \{x \in \mathbb{R}^3 : 0.15 \leqslant x_i \leqslant 1, i \in [3]\}$. For each polyhedron, we define a multivariate bandit instance with means and feasible region depicted in Figure 5.

We report in Table 4 the sample complexity of e-cAPE against that of the two-stage baseline and uniform sampling. We see that by focusing on responses with minimal cost, e-cAPE consistently achieves superior performance in our experiments. In some experiments, the uniform sampling approach combined with our stopping rule outperformed the two-stage algorithm. This outcome is anticipated, as our stopping rule is designed to trigger once any confidently correct answer can be identified. These findings underscore the competitiveness of both our sampling and stopping rules.

**Table 4:** Sample complexity of the algorithms averaged over 500 seeded runs.

| Experiment | Algorithms' empirical sample complexity | | | | | |
| | e-cAPE | | MD-APT+APE | | Uniform | |
| | Mean | Std | Mean | Std | Mean | Std |
|---|---|---|---|---|---|---|
| Simplex | **2679** | 517 | 4886 | 873 | 12156 | 2428 |
| Hypercube | **19034** | 3158 | 59841 | 10186 | 56184 | 14392 |
| Ordered polyhedron | **4313** | 693 | 7183 | 792 | 11895 | 1932 |

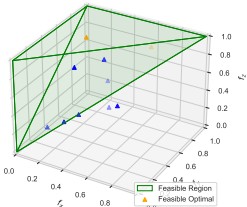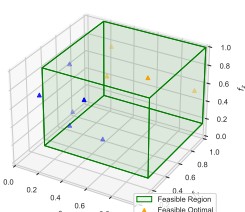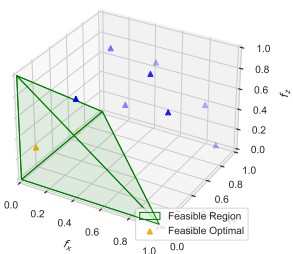

**Figure 5:** Constrained PSI instances with (left to right): ordered polyhedron, cube, simplex. The green area denotes the feasible region.

**Time and memory complexity of e-cAPE**    Since only the empirical means of the sampled arms $b_t, c_t$ change in each round, the Pareto set can be updated in time $O(Kd)$ and only the feasibility gaps of $b_t, c_t$ are updated at round $t$. We recall that $\text{dist}(\hat{\mu}_{t,i}, \partial P)$ has closed-form and computing $\text{dist}(\hat{\mu}_{t,i}, P)$ is a quadratic program that can be solved efficiently with libraries like CVXOPT. The memory complexity is $O(K^2)$, allocated only at initialization to store $M(i, j; t)_{i,j \in [K]}$. In Table 5 we report the runtimes of e-APE compared to that of the two-staged baseline and of uniform sampling.

**Table 5:** Runtimes of the algorithms averaged over 10 runs on the same instance.

| Experiment | Algorithms' runtimes (ms) | | | | | |
| --- | --- | --- | --- | --- | --- | --- |
| | **e-cAPE** | | MD-APT+APE | | Uniform | |
| | Mean | Std | Mean | Std | Mean | Std |
| Dose finding (Figure 2) | 2759 | 31 | 54890 | 381 | 3787 | 24 |
| Hypercube | 12300 | 9 | 18000 | 2970 | 2000 | 28 |
| Simplex | 672 | 13 | 2890 | 52 | 7543 | 57 |
| Ordered Polyhedron | 4010 | 7 | 4460 | 20 | 10200 | 40 |

## G.2. The cost of explainability

In this experiment, we present the result of a preliminary experiment to "compare" e-cAPE with Game-cPSI (Algorithm 2) our asymptotically optimal for the cPSI objective.

The two algorithms are not designed for the same objectives, but we can start to compare them for the simplest cPSI one, by evaluating e-cAPE only on its final guess $\widehat{O}_\tau$ for $O^*(\boldsymbol{\mu})$. Since e-cAPE is designed for the explainable case, it is expected to have a larger sample complexity. We evaluate both algorithms on a 5-armed Gaussian instance in dimension 2 with means given in Figure 6 (left). We set $\delta = 0.01$ and we report in Figure 6 (right) the distribution of the sample complexity over 250 seeded runs.

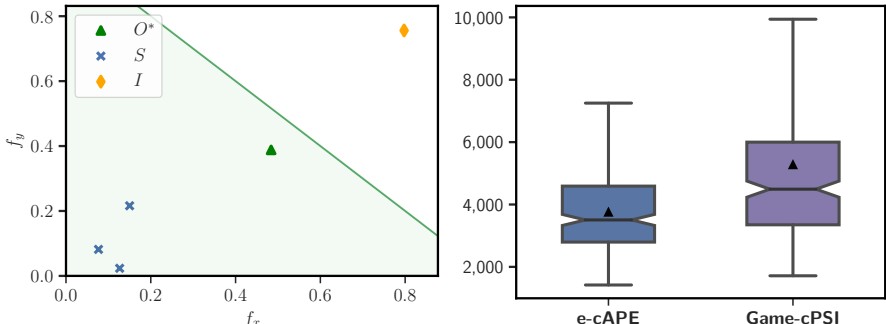

**Figure 6:** Constrained PSI instance (left) and distribution of the empirical sample complexity (right) averaged over 250 runs.

**Sample complexity**    Quite surprisingly, on that instance the sample complexity of e-cPSI is actually (slightly) smaller than of cPSI, despite the fact that the latter is asymptotically optimal for cPSI. There could be several explanations to this phenomenon: the performance of cPSI could be not so great in the moderate confidence regime, that is, for a fixed value of $\delta$ not tending to zero. Or, on this particular instance, the complexities of cPSI and e-cPSI could be close. This phenomenon will be investigated in future work by considering a larger benchmark.

**Error probabilities**    Additionally to the sample complexity, it is interesting to look at the empirical error, for both the cPSI and the e-cPSI objective. We let $\hat{\delta}_1$ denote the empirical correctness for the e-cPSI problem and $\hat{\delta}_2$ for the cPSI problem. By definition, we always have $\hat{\delta}_1 \leqslant \hat{\delta}_2$. Although Game-cPSI is not designed for e-cPSI we can use its empirical infeasible set and empirical feasible sub-optimal to form a complete recommendation for e-cPSI. For this heuristic variant of Game-cPSI, we have reported $(\hat{\delta}_1, \hat{\delta}_2) = (0.39, 0)$ while for e-cPSI we have reported $(\hat{\delta}_1, \hat{\delta}_2) = (0.01, 0)$. We remark that both algorithms are correct for their respective task. e-cPSI is by design also correct for cPSI, while the heuristic

version of Game-cPSI does not seem $\delta$-correct for e-cPSI, as suggested by the empirical results. This suggest that there exist an intermediate regime between cPSI and e-cPSI where the learner identifies the feasible Pareto Set before classifying the remaining arms.

### G.3. A Hard Instance for Racing

In this experiment, we illustrate the limitations of Racing algorithms for constrained PSI as argued in Appendix F. We run the experiment with $K = 5$ and $K = 10$ in dimension 2 on Bernoulli instances. In both cases, we keep the same polyhedron, which corresponds to a half-space delimited by the y axis. The instance is chosen such that each arm is feasible and there is only one optimal arm. We set $\delta = 0.01$ and we reported a negligible error. Each experiment is averaged over 250 runs.

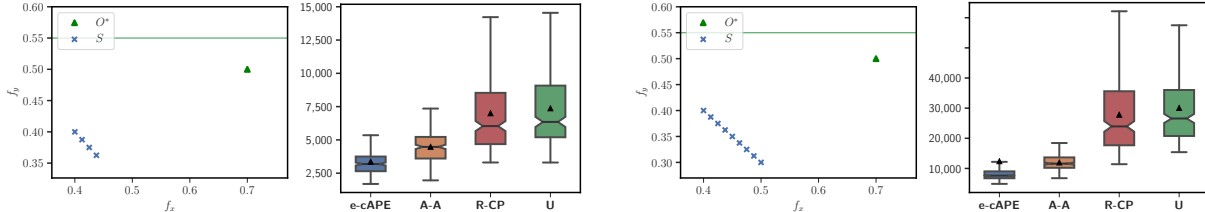

**Figure 7:** Constrained 5-armed PSI instance on the left and empirical distribution of the sample complexity (right).

**Figure 8:** Constrained 10-armed PSI instance on the left and empirical distribution of the sample complexity (right).

First, we mention that in this kind of instance, both e-cAPE and the two-stage approach have a closed theoretical sample complexity (in particular because the feasible sub-optimal arms are far from the borders). The results of Figure 8 show in this regime, Algorithm 2 is largely outperformed by e-cAPE as expected. Even the two-stage algorithm outperforms Algorithm 2 in this regime.

