# OpenReview forum: "Constrained Pareto Set Identification with Bandit Feedback"
_ICML.cc/2025/Conference — ICML 2025 poster_

### Official Review · Reviewer_meBs · 2025-03-10

**Overall Recommendation:** 3

**Summary:**

This paper studies the fixed-confidence setting of the Pareto Set under linear feasibility constraints in a multi-objective bandit setting. The authors propose an algorithm and establish its near-optimal theoretical guarantees through information-theoretic lower bounds in the worst case, and validate their approach through extensive experiments.

**Claims And Evidence:**

I did not identify any errors in the theoretical claims presented. However, I discuss certain limitations of these theoretical results in  'Other Strengths and Weaknesses.'

 Regarding the experimental claims, please refer to my detailed comments provided in the 'Questions for Authors' section.

**Essential References Not Discussed:**

I do not identify Essential References Not Discussed.

**Experimental Designs Or Analyses:**

The authors indicate that they conducted 500 runs for the experiments presented in Figure 4, and 250 runs for those in Figure 6. Could the authors also report the exact number of successful runs achieved by the agent in e-cPSI (or cPSI), and clarify whether these results align with the specified thresholds of $\delta = 0.1$ (Figure 4) and $\delta = 0.01$ (Figure 6)? Furthermore, to ensure a fair comparison, could the authors confirm whether the number of failures for e-cPSI (or cPSI) is approximately consistent across the evaluated algorithms in these experiments?"

**Methods And Evaluation Criteria:**

In the experiment, it makes sense to compare the empirical stopping time. However, it would be better to also show the empirical  failure rate.

**Other Comments Or Suggestions:**

Line 85: Algorithm XX is a typo to be fixed.

**Other Strengths And Weaknesses:**

## Strengths:
In my view, the key contributions of this paper are captured by Theorems 4.3 and 4.4. Specifically, Theorem 4.3 establishes a non-asymptotic upper bound, while Theorem 4.4 provides a corresponding lower bound, which is particularly valuable as it matches the upper bound of Theorem 4.3, albeit in a worst-case scenario.


## Weaknesses:
1. The motivation behind introducing linear constraints is not sufficiently clear. Although the authors provide an illustrative example concerning clinical trials, this example lacks detailed specifics to fully justify the necessity and relevance of incorporating linear constraints.
2. The practical applicability of the proposed algorithms appears significantly limited regarding the parameter $\delta$. According to Theorem 4.3, the algorithm is guaranteed to be $\delta$-correct only when $\delta < d^2/5^d$. For instance, setting $d=10$ imposes a condition $\delta < 10^{-5}$, severely restricting the range of realistic scenarios where this method could be applied effectively.
3. Within the main text, there is a noticeable absence of theoretical analysis specifically addressing the cPSI performance of the proposed algorithm. It would be beneficial to clarify whether the presented algorithm achieves near-optimality in terms of cPSI, or it needs a significant modification on the proposed algorithm to adapt the task of cPSI.

**Questions For Authors:**

1. In Section 5, the authors propose a two-stage algorithm comprising feasible set identification and Pareto set identification. How is the confidence level $\delta$ configured in each of these two stages? Additionally, is historical data from the first stage utilized in the second stage?

2. Could the authors clarify the meaning of the phrase "until a correct set is identified with high probability" in the context of the baseline method labeled "Uniform"?

3. The authors state that the experiments depicted in Figure 4 were run 500 times. Could they also report how many times the agent succeeded in achieving e-cPSI, and whether this empirical success rate aligns with the theoretical confidence level of $\delta = 0.1$? Furthermore, for a fair comparison, are the failure rates approximately consistent across different algorithms?

4. What is the precise relationship between $T^*_{\mathcal{M}}$ and $C^*_{\mathcal{M}}$ in the worst-case scenario discussed in Theorem 4.4?

5. What is the computational complexity of the proposed algorithm?

**Relation To Broader Scientific Literature:**

The primary contribution of this paper lies within the domain of multi-objective multi-armed bandits. While most prior studies have concentrated on regret minimization or unconstrained Pareto set identification, this paper presents the first investigation into constrained Pareto set identification, even within the simpler setting of linear constraints.

**Theoretical Claims:**

I do not check the proof.

---

> ### Author Rebuttal · Authors · 2025-04-01
>
> We appreciate the reviewer's thoughtful feedback and detailed evaluation of our work. Below, we address each point raised by the reviewer.
> - Weaknesses
>   1. Linear constraints are widely used in applications like dose-finding trials, where balancing efficacy and toxicity is crucial (Mark C et al., 2013). Appx G.2 (Tab. 5) includes additional results on more complex set of linear constraints. Our guarantees also extend to general convex constraints, but we focus on polyhedral ones for their practicality and efficiency.
>   2. We would like to clarify that this condition was actually needed only for the sample complexity bound and not for the correctness (for any $δ$, e-cAPE outputs a correct answer with probability larger than $1−δ$). We acknowledge that this was unclear from our statement of Thm 4.3. This condition is actually loose and can be removed (at the cost of a second-order term in the sample complexity independent from $δ$).
>   3. e-cAPE is mainly designed e-cPSI, but any $𝛿$-correct algorithm for e-cPSI is also $𝛿$-correct for cPSI; so is e-cAPE. However, as both tasks are different, e-cAPE is not necessarily optimal for cPSI. As noted in Section 3.1, we provide a dedicated algorithm for cPSI in Appx E, which is optimal as $𝛿\to 0$. An experiment in Appx G.2 compares e-cAPE to this cPSI-specific algorithm for $𝛿=0.01$. e-cAPE performed slightly better in practice despite not being tailored for cPSI. If given additional space in a revision, we would include more details on cPSI in the main text to further clarify its theoretical and practical aspects.
>
> - **Questions**
>   1. We set $𝛿/2$ for each stage to maintain an overall $𝛿$-correct guarantee. We did not retain historical data from the first stage (feasibility identification) to the second stage (Pareto set identification) as it is unclear whether the claimed sampled complexity bound holds if we do. Through experiments on synthetic instances, we actually observed that reusing historical data from the first stage often introduced bias in the second stage, leading to an overall increase in sample complexity. Exploring more principled ways to leverage historical data while mitigating bias is an interesting direction for future work.
>   2. By this, we mean that the algorithm runs until the stopping condition specified in Equation (7) is met. We clarify that the "Uniform" baseline shares the same stopping criterion as e-cAPE; the key difference lies in their sampling strategies. Specifically, while e-cAPE adaptively allocates samples, the "Uniform" algorithm follows a round-robin sampling scheme, allocating roughly equal samples to all arms.
>
>   3. For $𝛿=0.1$, we report below the empirical success rate of each algorithm
> ||e-cAPE|Uniform(U)|MD-APT+APE(A-A,Two Stage)|Racing algorithm(R-CP)|
> |---|---|---|---|---|
> |covboost|100%|100%|100%|96%|
> |secukinumab|100%|100%|100%|100%|
>
> All algorithms achieve similar success rates, and e-cAPE, R-CP, A-A, implement the same confidence bonus function. "Uniform" and e-cAPE share the same stopping rule. As detailed in Appx G, in the implementation, we used the confidence thresholds advertised in Auer et al (2016). Although tighter than the one allowed by the theory, they remain very conservative.
>
>   4. $T^*_{M}$ captures the hardness of e-cPSI in the asymptotic (when $𝛿\to 0$, cf Prop 3.4). As noted in section 3.2, an extension of Degenne and Koolean (2019) to e-cPSI would yield an optimal (as $𝛿\to 0$) yet impractical algorithm (as it needs to solve up to $2^K$ max-min problems similar to Eq.(2) at each round, none having a closed form). This algorithm will satisfy $\lim_{𝛿\to 0}\frac{\mathbb{E}_{{\mu}}[τ_𝛿]}{\log(1/𝛿)}=T^*\_{M}$. On the other side, $C_M^*$ is the leading complexity term in our lower bound and in the sample complexity of e-cAPE, for which the guarantees are non-asymptotic. By combining Thm 4.3 and 4.4, on the class of problems $\tilde{D}$ (explicit in appx B), we have
>   $$C_M^*(\mu)/4⩽ T^*_M(\mu)⩽256C_M^*(\mu),$$
>  which shows that $C_M^*(\mu)$ is a reasonable complexity proxy for problems in $\tilde{D}$ (up to some improvable constants).
>
>   5. The computational complexity of e-cAPE is mainly determined by computing the squared Euclidean distance $dist(x,P)^2$, which we solve using MOSEK (via CVXOPT) in $O(\max(m,d)^3)$ with $d$ the dimension and $m$ the number of constraints. Computing $dist(x,P^c)$ takes $O(md)$. If the state (quantities $M(i,j;t)^\pm_{i,j},(γ_i(t))_{i}$,) is updated, $b_t,c_t$ are computed in $O(K)$. As only the means of $b_t,c_t$ change from $t$ to $t+1$, updating the state requires :
> * $O(n_t d+md)$ to update the feasible Pareto set ($n_t$: size of the feasible set).
> * $O(K)$ to update $M(i,j)^\pm$
> * $O(\max(m,d)^3+\max(K,m)d)$ if $b_t$ or $c_t$ is empirically infeasible, otherwise $O(\max(K,m)d)$ to update $(γ_i(t))$.
>
> Per iteration, the cost is linear when $b_t,c_t$ are feasible; otherwise, cubic in $m$. Memory complexity is $O(K^2)$ to store the state.

---

> > ### Comment · Reviewer_meBs · 2025-04-05
> >
> > The experiments could be better, as the $100%$ success rate is too conservative for a fair comparison. However, considering this is a theoretical paper, I am satisfied with the overall response and have increased my score from 2 to 3. I would encourage the authors to incorporate the above discussion into the revised paper.

---

> > > ### Author Response · Authors · 2025-04-06
> > >
> > > We thank the reviewer again for taking the time to review our paper and engage with our rebuttal. We are pleased that the reviewer found our clarifications helpful and sincerely appreciate the updated score and thoughtful feedback.  We also take note of the reviewer’s comments on the experimental setup and will incorporate the suggested discussion into the revised version.

---

### Official Review · Reviewer_DwcZ · 2025-03-12

**Overall Recommendation:** 3

**Summary:**

This paper studies the constrained Pareto set identification (cPSI) problem (with explainability). More specifically, the authors focus on the $(\epsilon, \delta)$-PAC learning setting, and the objective of the agent is to identify a partition of the arm set into three sets (the Pareto set, a set of suboptimal arms, a set of infeasible arms). They derived instance-dependent lower bounds of the sample complexity for cPSI and e-cPSI. Then, they proposed an algorithm for e-cPSI termed e-cAPE. They provide the sample complexity analysis of e-cAPE and prove that it is nearly optimal for some problem instances, which implies that e-cAPE is nearly optimal in a worst-case sense. Using real-world datasets, they empirically show that the proposed method achieves smaller sample complexities for some problem instances.

## update after rebuttal
Since my concerns (questions) have been resolved by the author's response, I will keep my positive score.

**Claims And Evidence:**

Claims are supported by theoretical results (analysis on lower and upper bounds) and experimental results.

**Essential References Not Discussed:**

To the best of my knowledge, related works are adequately discussed.

**Experimental Designs Or Analyses:**

The experiments are conducted using real-world datasets and baselines seem valid.

**Methods And Evaluation Criteria:**

Theorem 4.4 shows the proposed method is nearly optimal for some problem instances and the evaluation metric for experiments is the sample complexity, which is standard for the PAC problem.

**Other Comments Or Suggestions:**

Is an algorithm for cPSI asymptotically optimal? If so, since the current version of this manuscript has some space, you can include the claim even in the submitted version (and a revision).

**Other Strengths And Weaknesses:**

- Strengths
    - The authors study practically important problem settings (cPSI and e-cPSI).
    - Experiments are conducted on real-world datasets.
    - The proposed method is near-optimal in a worst-case sense.
- Weaknesses
    - Although the proposed method is near-optimal for some problem instances, it is a worst-case analysis. I suspect, a very simple algorithm (such as uniform exploration) has a similar property.

**Questions For Authors:**

1. As I wrote in "weaknesses", I think a uniform exploration algorithm can achieve an optimality in a worst-case. Is it possible to theoretically compare Algorithm 1 to such a simple algorithm?

**Relation To Broader Scientific Literature:**

As the authors state cPSI and e-cPSI are practically important problem settings, such algorithms would be beneficial to the papers outside the ML community.

**Theoretical Claims:**

I have not checked proofs. At least, lower bounds seem standard results and valid.

---

> ### Author Rebuttal · Authors · 2025-03-31
>
> We thank the reviewer for their comments and for taking the time to evaluate our work. Below, we address each of the points raised in the review.
>
> * **Optimal algorithm for cPSI**
>
> We present an asymptotically optimal algorithm for cPSI in Appendix E. Since we believed that e-cPSI was better suited for the applications we focused on, we initially placed the cPSI algorithm in the appendix. However, as suggested by the reviewer, we can provide more details on this algorithm in the main text if space allows in the revision.
>
> * **Uniform exploration**
>
> We agree that an algorithm using uniform exploration could be worse-case optimal for certain very particular configurations of arms (but probably a much more reduced set of instances compared to the one constructed in our lower bound). Still, e-cAPE is designed to perform more efficiently by focusing on arms that are more likely to be part of the feasible Pareto Set rather than exploring uniformly. This adaptive exploration typically leads to more efficient identification of the Pareto Set, especially when the number of arms is large.
> In Appendix F, we analyze the limitations of an algorithm that performs uniform exploration and additionally discards some arms when their status can be deduced from confidence boxes. We show that for this algorithm (which is expected to be even better than pure uniform sampling due to the additional eliminations), there are some configurations where its sample complexity scales with a quantity that is order $K$ times larger than that of e-cAPE; with $K$ the number of arms. This provides some theoretical insights as to why the sampling rule of e-cAPE leads to a lower sample complexity compared to uniform sampling.
> Moreover, we do illustrate empirically the benefits of e-APE compared to both uniform sampling (see e.g. Table 5 in Appendix G.2) and uniform sampling with eliminations (see Figure 6,7 in Appendix G.2).

---

> > ### Comment · Reviewer_DwcZ · 2025-04-08
> >
> > I thank the authors for their detailed response. My question on uniform exploration has been resolved. I will keep my current score.

---

### Official Review · Reviewer_7kAg · 2025-03-12

**Overall Recommendation:** 3

**Summary:**

This work studies bandit pareto set identification with constraints. In particular, the task is to choose arms to pull in each round until the set of feasible and pareto arms is identified with probability at least $1 - \delta$. They give an algorithm that addresses this problem

## update after rebuttal

In the rebuttal, the authors clarified the meaning of $\lambda_\alpha$, and provided an additional result that doesn't require that $\delta$ shrinks exponentially in $d$. Regarding the clarification of $\lambda_\alpha$, I found the authors' explanation to be sound and quite helpful in general. Regarding the additional result, I do think that this alleviates some of the restrictions of the original result. However, one downside of this result is that we can't say much about the tightness (although this is not true in the reasonable setting where $\delta$ is small). Overall, I maintain my original score, as I think the results are good overall, but there are some points of weakness.

**Claims And Evidence:**

I have some concerns with the interpretation of the sample complexity bound. In particular, I'm not sure the claim of "near-optimal" is justified given my concerns detailed in \# 1 and \# 2 in the Questions box.

**Essential References Not Discussed:**

None that I saw.

**Experimental Designs Or Analyses:**

There are no experiments.

**Methods And Evaluation Criteria:**

Overall, the approach seams reasonable, and the use of sample complexity to evaluate the algorithm is reasonable.

**Other Comments Or Suggestions:**

1. I would suggest putting a definition of the Pareto optimal set $O^*$ in Section 1.1. It would be preferable for this to appear before it is referenced in line 97.
2. There is an unfilled reference XX in line 85 right side.

**Other Strengths And Weaknesses:**

No others.

**Questions For Authors:**

1. The sample complexity guarantees in Theorem 4.3 are shown to include a term $\lambda_\alpha$ which looks to be $\tilde{O}(\sum_{T \geq 1} \frac{1}{T^{\alpha-1}})$ where $\alpha$ is a free quantity restricted to $\alpha > 2$. It seems that the claimed regret bounds would require that $\lambda_\alpha = \tilde{O}(1)$, but I don't see how this would be the case when the factor $\sum_{T \geq 1} \frac{1}{T^{\alpha-1}}$ could be as large as $\sum_{T \geq 1} \frac{1}{T}$. Maybe it is just not clear to me what the summation $\sum_{T \geq 1}$ is over exactly.
2. The sample complexity guarantees in Theorem 4.3 also restrict the confidence level $\delta$ to be in the range $\delta \leq \frac{d^2}{5^d}$, which depends exponentially on the dimension. This seems to be highly restrictive, especially in high dimension settings. Can this be avoided? I didn't see this requirement in related work.
3. Can the approach be extended to convex sets? I did not identify any specific arguments that restricted the algorithm or analysis to polytopes.

**Relation To Broader Scientific Literature:**

To my knowledge, this works contributes a new problem setting to the literature, combining the pareto set identification problem with the feasible set identification problem. Accordingly, their approach appears to be novel.

**Theoretical Claims:**

I did not check the proofs, but have some concerns with the interpretation of the theoretical results as discussed in the Question box \# 1 and \# 2.

---

> ### Author Rebuttal · Authors · 2025-04-01
>
> We appreciate the reviewer's thoughtful feedback and detailed evaluation of our work. Below, we specifically address each point raised in the review.
>
> 1.  $α$ is a parameter of the algorithm. We introduced generic confidence bonuses in l.255-l.257, and Thm 4.3 upper-bounds the sample complexity of e-cAPE when it is run with confidence bonuses in the form described in the statement (depending on $α$). In practice, $\alpha$ is set at the beginning of the algorithm. We give additional clarification on $Λ_α$ in the answer to question 2 below.
>
> 2. We thank the reviewer for this insightful comment. The condition on $δ$ is actually loose and can be removed (at the cost of an additional second-order term, independent from $δ$). First, we would like to mention that this condition was actually needed only for the sample complexity bound and not for the correctness of the algorithm (e-cAPE can be run for any $δ$ and outputs a correct answer with probability larger than $1−δ$). We acknowledge that this is unclear from our current statement of Thm 4.3.
> The term  $5^d$ that appears in the calibration function $g(t,δ)$ is due to the covering number of the unit sphere, which arises from L2 norm concentration with the covering technique. The condition on $δ$ appeared from a sub-optimal upper bound on $g(t,δ)$ in the proof, specifically in l.1300. To see how this can be improved, we sketch below the proof of Thm 4.3
>
>   * **Bounding the stopping time under consecutive good events**
>   Let $τ$ be the (random) stopping time of e-cAPE, and observe that for any $T>0$,
> $$\min(τ,T)⩽T/2+\sum_{t=T/2}^T\mathbb{1}_{(τ>t)}.$$
>
> In Proposition C.3, we show that if the good event $E_{t}$ (defined in l.255-256) holds and e-cAPE does not stop at round
>
> $t$, then for any correct answer $(S,I)\in M$, either $b_t$ or $c_t$ is underexplored (i.e. not pulled enough wrt some gaps that are function of $S, I$). This is expressed by saying that $\\{b_t,c_t\\}\cap W_t(S,I)\neq ∅$; $W_t(S,I)$ is defined in equations 37-39 (l.980-83). Assuming $$E^T:=\bigcap_{\frac{T}{2}⩽ t⩽T}E_t$$ holds and fixing an arbitrary correct answer $(S,I)$, we have
>
> $$ \min(τ,T) ⩽ T/2+\sum_{a=1}^K\sum_{t=T/2}^T\mathbb{1}_{((b_t=a\lor c_t=a)) \land (a\in W_t(S,I))}.$$
>
> Introducing for any subset $U\subset[K]$,
>
> $$R(U,T) :=\sum_{a\in U}\sum_{t=T/2}^T\mathbb{1}_{((b_t=a\lor c_t=a)) \land (a\in W_t(S,I))},$$
>
> it follows from the definition of $W_t(S,I)$ (equation 37-39, l.980-83) that
> $$R(S,T)⩽\sum_{a\in S}\frac{32σ^2}{\Delta_a^{2}(S)}f(T,𝛿);R(I,T)⩽\sum_{a\in I}\frac{8σ^2}{\eta_a^2}g(T,𝛿)$$ and
>  $$R(O^*,T)⩽\sum_{a\in O^*}σ^2\max(\frac{8g(T,𝛿)}{\eta_a^2},\frac{32f(T,𝛿)}{Δ_a^2(S)}).$$
>    * **Former sub-optimal step and novel formulation of Thm 4.3**
>
> At this step, the idea was to write $R(O^\star)$ a sum of terms in the form $\sum_{a\in O^*}\max(\frac{1}{Δ_a^2(S)},\frac{1}{\eta_a^2})h(T,𝛿)$ (this would further make appear the complexity term $C(S,I)$, cf Eq.11 (l.311)).
> Recalling $$f(T,𝛿)=\log(\frac{2k_1KdT^α}{𝛿})\text{ and }g(T,𝛿)=4\log(\frac{2k_1K5^dT^α}{𝛿}),$$
> this is where we used the condition: as for $𝛿⩽ d^2/5^d$ we have
> $\log(5^d/𝛿)⩽2\log(d/𝛿)$ and $g(T,𝛿)⩽ 8f(T,𝛿)$; and we set $h(T,𝛿)=8f(T,𝛿)$. To fix this loose step and remove the restriction on $𝛿$, observe that
> $$R(O^*,T)⩽\sum_{a\in O^*}32σ^2\max(\frac{1}{\eta_a^2},\frac{1}{Δ_a^2(S)})f(T,𝛿)+Q(\mu,O^*)$$
> where $Q(\mu,U)=\sum_{a\in U}32σ^2\frac{\log(5^d/d)}{\eta_a^2}$.
> Applying the modification above and following the remaining proof of Thm 4.3, we prove that for any $𝛿\in(0,1)$,
> $$\mathbb{E}[τ]⩽256σ^2 C^*(\mu)\log(128σ^2C^*(\mu)(2k_1Kd/𝛿)^{1/α})+4Q(\mu,F^c \cup O^*)+Λ_α$$
> where $F$ is the feasible set.
>
>  * **Additional Clarification on $Λ_α$**
>
> By definition, we have (l.1330) $Λ_α=\sum_{T⩾ 1}\mathbb{P}((E^T)^c)$. We showed above that when $E^T$ holds,
> $$\min(τ,T)⩽ T/2+R([K],T)$$ then, letting $\tilde T$ such that $\forall T⩾\tilde T,R([K],T)<T/2$,
> for $T⩾\tilde T$, $τ >T⟹ (E^T)^c$. Thus,
> $$\mathbb{E}[τ]⩽\tilde T+\sum_{T⩾\tilde T}\mathbb{P}((E^T)^c)⩽\tilde T+Λ_α,$$ and $Λ_α$ is bounded using Lem D.3 in appendix.
>
> 3. We appreciate the reviewer's insightful observation. Indeed, our approach could be extended to general convex feasible sets. However, the main challenge would be computational. Specifically, for a feasible set $P$, our algorithm should compute at each iteration, quantities such as $dist(x,P)^2$ (squared Euclidean distance to $P$; convex quadratic program with linear constraints) and $dist(x,P^c)^2$, which is not always convex, making it significantly more complex in general. In the special case of polyhedral sets, the latter distance has a closed-form expression, simplifying computations considerably. Additionally, another computational aspect (albeit minor) is the verification oracle for membership in $P$. Given that polyhedral sets encompass many practical scenarios while ensuring efficient algorithmic costs, we chose to focus the presentation on this setting.

---

> > ### Comment · Reviewer_7kAg · 2025-04-03
> >
> > Thank you for the detailed discussion and walking me through the proof steps. I found this alternative proof method convincing and have a clearer picture on the results in general. Is there anything that we can say about the tightness of this modified bound? The additional term is something like $d \sum_a \frac{1}{\eta_a^2}$. This seems to be incomparable to $C_M^*$.

---

> > > ### Author Response · Authors · 2025-04-04
> > >
> > > We thank the reviewer for the follow-up and are glad the clarification helped. Regarding the additional term $d \sum_{a \in O^\star \cup F^c} \frac{1}{\eta_a^2}$, we agree that in general it is not directly comparable to $C_M^\star$ which involves gaps tied to Pareto set identification.
> > > However as it is $\delta$-independent, when $\delta$ is small the leading term will be $C_M^\star$, which is also the term featured in our lower bound.
> > >
> > > The tightness of this second-order term depends directly on the tightness of the L2 norm concentration (i.e. on the choice of the confidence function $g(t,\delta)$). With standard covering arguments, it scales with the covering number of the unit sphere, which is exponential in dimension.
> > >
> > > This can be improved under stronger assumptions on the distributions of the arms. While our analysis assumes marginal-wise $\sigma$-sub-Gaussianity (a standard assumption in multi-objective bandits, e.g., Auer et al. 2016), tighter bounds (for smaller calibration functions $g(t,\delta)$) can be obtained under stronger assumptions:
> > > * **multivariate Gaussian with known covariance**: if we assume each arm to be multivariate Gaussian with known covariance, using Hanson-Wright concentration for Gaussian vectors (see (Rudelson and Vershynin 2013, Hanson-Wright inequality and sub-gaussian concentration) ) will improve the dependency in the log from exponential in the dimension to the operator norm of the covariance matrix.
> > >
> > > * **Independent marginals** : Kaufmann and Koolen 2021 (in Mixture Martingales Revisited with Applications to Sequential Tests and Confidence Intervals) provides refined concentration bounds on the sum of KL divergences from which tighter L2-norm concentration can be deduced for random vectors with independent sub-gaussian marginals. Using their results, we could take state a high-probability bound on the sample complexity of the resulting algorithm (instead of the expected sample complexity we bound here) where the second-order term will be in $\log\log(d)$.
> > >
> > > Tightening the second-order term under realistic assumptions on the arms is an interesting direction that we plan to explore in future work.

---

### Decision · Program_Chairs · 2025-05-01

**Decision:**

Accept (poster)

**Comment:**

The paper investigates bandit Pareto set identification problems under constraints, where each arm corresponds to a distribution in $\mathbb{R}^d$. The goal is to identify the set of Pareto-optimal arms whose mean vectors lie within a known polyhedron. The authors study this problem in the fixed-confidence setting, deriving an instance-specific lower bound on the sample complexity and proposing near-optimal, computationally efficient algorithms. They also present numerical experiments inspired by real-world data.

The techniques used in the paper are fairly standard, but the reviewers are generally positive. The authors have adequately addressed all concerns raised during the discussion phase.

The paper could be further improved by enhancing the motivation in the introduction—for example, clarifying the significance of the problem and explaining the relevance of using polyhedral constraints.